# Identification and characterization of diverse OTU deubiquitinases in bacteria

Alexander F Schubert[1,‡,†] [ID], Justine V Nguyen[2,†] [ID], Tyler G Franklin[2] [ID], Paul P Geurink[3] [ID], Cameron G Roberts[2] [ID], Daniel J Sanderson[2] [ID], Lauren N Miller[2] [ID], Huib Ovaa[3] [ID], Kay Hofmann[4] [ID], Jonathan N Pruneda[1,2,*] [ID] & David Komander[1,5,6,**] [ID]

## Abstract

Manipulation of host ubiquitin signaling is becoming an increasingly apparent evolutionary strategy among bacterial and viral pathogens. By removing host ubiquitin signals, for example, invading pathogens can inactivate immune response pathways and evade detection. The ovarian tumor (OTU) family of deubiquitinases regulates diverse ubiquitin signals in humans. Viral pathogens have also extensively co-opted the OTU fold to subvert host signaling, but the extent to which bacteria utilize the OTU fold was unknown. We have predicted and validated a set of OTU deubiquitinases encoded by several classes of pathogenic bacteria. Biochemical assays highlight the ubiquitin and polyubiquitin linkage specificities of these bacterial deubiquitinases. By determining the ubiquitin-bound structures of two examples, we demonstrate the novel strategies that have evolved to both thread an OTU fold and recognize a ubiquitin substrate. With these new examples, we perform the first cross-kingdom structural analysis of the OTU fold that highlights commonalities among distantly related OTU deubiquitinases.

**Keywords** bacterial effector; deubiquitinase; pathogen; protein structure; ubiquitin
**Subject Categories** Microbiology, Virology & Host Pathogen Interaction; Post-translational Modifications & Proteolysis; Structural Biology
**The EMBO Journal (2020) 39: e105127**

## Introduction

Outside of its canonical role in targeted proteasomal degradation, ubiquitin (Ub) signaling plays crucial roles in many other aspects of eukaryotic biology, including immune responses (Swatek & Komander, 2016; Ebner *et al*, 2017). In fact, the ability of Ub modifications to form discrete polymers (polyUb) allows it to perform multiple signaling functions even within the same pathway (Komander & Rape, 2012). TNF signaling, for example, relies upon the concerted action of several nondegradative polyUb signals (K63-, M1-, and K11-linked chains) as well as the degradative K48-linked chains in order to ultimately achieve NF-κB transcriptional activation (Ebner *et al*, 2017). PolyUb chains can also be combined into complex higher-order architectures that further diversify their signaling capacities (Haakonsen & Rape, 2019). These processes are tightly regulated by Ub ligases that assemble the signals, Ub-binding domains that respond to them, and specialized proteases termed deubiquitinases (DUBs) that remove them. Breakdown of this regulation can lead to immune hyper- or hypoactivation, and has been linked to several human diseases (Popovic *et al*, 2014).

Although the Ub system is largely exclusive to eukaryotes, invading viruses and bacteria have evolved strategies for manipulating host Ub signaling responses during infection (Wimmer & Schreiner, 2015; Lin & Machner, 2017). These strategies can include pathogen-encoded Ub ligases or DUBs that redirect or remove host signals, respectively. Pathogen-encoded DUBs can affect host functions such as innate immune activation, autophagy, or morphology (Mesquita *et al*, 2012; Pruneda *et al*, 2018; Wan *et al*, 2019). When their ability to remove host Ub signals is taken away, some pathogens show reduced fitness and infectivity (Rytkönen *et al*, 2007; Fischer *et al*, 2017). Interestingly, though some bacterial DUBs are entirely foreign and reflect convergent evolution (Wan *et al*, 2019), others appear to adopt eukaryote-like protein folds and/or mechanisms (Pruneda *et al*, 2016).

Humans encode six families of cysteine-dependent DUBs that all fall underneath the CA clan of proteases and one family of Ub-specific metalloproteases from the MP clan. An additional family of

1   Medical Research Council Laboratory of Molecular Biology, Cambridge, UK
2   Department of Molecular Microbiology & Immunology, Oregon Health & Science University, Portland, OR, USA
3   Oncode Institute & Department of Cell and Chemical Biology, Leiden University Medical Centre, Leiden, The Netherlands
4   Institute for Genetics, University of Cologne, Cologne, Germany
5   Ubiquitin Signalling Division, The Walter and Eliza Hall Institute of Medical Research, Parkville, VIC, Australia
6   Department of Medical Biology, The University of Melbourne, Melbourne, VIC, Australia
    *Corresponding author. Tel: 1 (503) 494 8102; E-mail: pruneda@ohsu.edu
    **Corresponding author. Tel: +61 3 9345 2670; E-mail: dk@wehi.edu.au
    †These authors contributed equally to this work
    ‡Present address: Department of Structural Biology, Genentech Inc., South San Francisco, CA, USA

ubiquitin-like proteases (ULPs) regulates NEDD8 and SUMO signaling and belongs to the CE cysteine protease clan. The majority of bacterial DUBs studied to date are related to the CE clan of ULPs and appear to predominantely target host K63-linked polyUb signals (Pruneda *et al*, 2016). The ULP fold is also widely used among viruses, both as a Ub-specific protease and as a traditional peptidase (Wimmer & Schreiner, 2015).

Another DUB fold that is common to both eukaryotes and viruses is the ovarian tumor (OTU) family. Humans encode 16 DUBs of the OTU family with important functions in signaling pathways such as innate immunity and cell cycle regulation (Du *et al*, 2019). Some OTUs, such as OTUB1 and OTULIN, are highly specific for certain polyUb signals (K48- and M1-linked chains, respectively), and these properties not only provide insight into their biological functions (proteasomal degradation and inflammatory signaling, respectively), but also prove useful for technological applications such as ubiquitin chain restriction analysis (Keusekotten *et al*, 2013; Mevissen *et al*, 2013; Du *et al*, 2019). Viruses use OTU DUBs to block innate immune activation during infection, often by cleaving both Ub and the antiviral Ub-like modifier ISG15 (Bailey-Elkin *et al*, 2014). In bacteria, however, only two reported cases of the OTU fold have been identified. The first, ChlaOTU from *Chlamydia pneumoniae*, was predicted by sequence similarity (Makarova *et al*, 2000) and shown to play an active role in the clearance of Ub signals following infection (Furtado *et al*, 2013). The second example, LotA, plays a similar role in *Legionella pneumophila* infection (Kubori *et al*, 2018). Whether these bacterial OTUs were unique, however, or represent a wider adaptation of the OTU fold among bacteria remained unknown.

To determine whether, like the CE clan ULPs, the OTU fold is a common adaptation for DUB activity across bacteria, we generated an OTU sequence profile and predicted distantly related examples among bacterial genomes. Using an array of Ub substrates and *in vitro* assays, we confirmed that predicted OTUs from pathogens such as *Escherichia albertii*, *L. pneumophila,* and *Wolbachia pipientis* were *bona fide* DUBs. Furthermore, with one exception all of our confirmed OTUs were Ub-specific (over Ub-like modifiers) and targeted a defined subset of polyUb chain types, much like human OTUs (Mevissen *et al*, 2013). Structural analysis of two examples revealed novel modes of Ub substrate recognition and, surprisingly, even a permutated sequence topology that still gives rise to a familiar OTU fold. Our new bacterial OTU DUB structures allowed for the first cross-kingdom structural analysis, from which we established a framework for identifying evolutionary adaptations in the S1 substrate-binding site that impart DUB activity. This work establishes the OTU fold as a common tool used by bacteria to manipulate host Ub signaling and provides insight into the origins and adaptations of the OTU fold across eukaryotes, bacteria, and viruses.

## Results

### Identification of bacterially encoded OTU deubiquitinases

Given the expansive use of the OTU DUB fold in eukaryotes and viruses to regulate key aspects of cellular biology and infection, respectively (Du *et al*, 2019), we sought to determine whether, like

the CE clan ULPs (Pruneda *et al*, 2016), the family extends into bacteria as well. Through generating a sequence alignment of eukaryotic and viral OTU domains, we created a generalized sequence profile that was used to identify related sequences among bacteria. Candidates identified through this approach were further scrutinized by secondary structure prediction and domain recognition using the Phyre2 server (Kelley *et al*, 2015). Those that encoded active site sequences matching the Pfam motif (Pfam Entry PF02338) embedded within appropriate elements of secondary structure (e.g., an active site Cys motif at the beginning of an α-helix) were prioritized for subsequent validation (Fig EV1A). Reassuringly, this approach also detected the first characterized bacterial OTU, ChlaOTU (Makarova *et al*, 2000; Furtado *et al*, 2013), and we followed this naming convention for predictions with previously unknown function. Despite encoding two OTU domains (Kubori *et al*, 2018), LotA was not detected by our approach. For biochemical validation, we selected *E. albertii* "EschOTU" (GenBank EDS93808.1), *L. pneumophila* ceg7 (lpg0227, GenBank AAU26334.1), *Burkholderia ambifaria* "BurkOTU" (GenBank EDT05193.1), *C. pneumoniae* ChlaOTU (CPn_0483, GenBank AAD18623.1), *Rickettsia massiliae* "RickOTU" (dnaE2, GenBank ABV84894.1), *W. pipientis* strain wPip "wPipOTU" (WP0514, GenBank CAQ54622.1), *W. pipientis* wMel "wMelOTU" (WD_0443, GenBank AAS14166.1), and *L. pneumophila* ceg23 (lpg1621, GenBank AAU27701.1) (Fig 1A and B).

Our selected candidates are encoded by a wide range of Gram-negative bacteria that span the chlamydiae, alpha-, beta-, and gammaproteobacterial classes (Fig 1B). Consistent with putative host-targeted DUB activity, all of the identified species have reported interactions with eukaryotic hosts (Fig 1C), some of which are linked to severe human diseases (e.g., Legionnaire's disease) or altered biology (e.g., Wolbachia sex determination). The majority of our candidates arise from obligate intracellular bacteria that depend upon host interactions for survival. All of the selected bacterial OTU-containing proteins were predicted by pEFFECT or S4TE 2.0 to also encode either type III or type IV secretion signals (Goldberg *et al*, 2016; Noroy *et al*, 2019), suggesting potential roles as secreted effectors (Fig EV1B).

With the exception of ChlaOTU, which had no recognizable conservation of the general base His motif, all of the selected examples contained both catalytic Cys and general base His consensus sequences that closely matched the established motifs and secondary structure of OTUs (Fig 1A). Remarkably, however, our active site analysis suggested that some examples, particularly EschOTU, could thread through the OTU fold in a topology that is distinct from any previously studied example (Fig 1A, red arrow). Outside of the active site motifs, our OTU domain predictions have strikingly low sequence similarity to each other and to the archetypal human example, OTUB1, that centers around only ~ 15% identity (Fig 1D).

To test our predictions for DUB activity, we synthesized coding regions or amplified them from bacterial samples, designed constructs that (where possible) contain the minimal predicted OTU domain, and proceeded with *Escherichia coli* expression and purification (Fig 1E). We found the *Legionella* ceg7 protein to be the most difficult to work with, and after much effort arrived at a preparation that retained a SUMO solubility tag (Fig 1E). As a first measure of *in vitro* DUB activity, we treated the putative bacterial OTUs with a

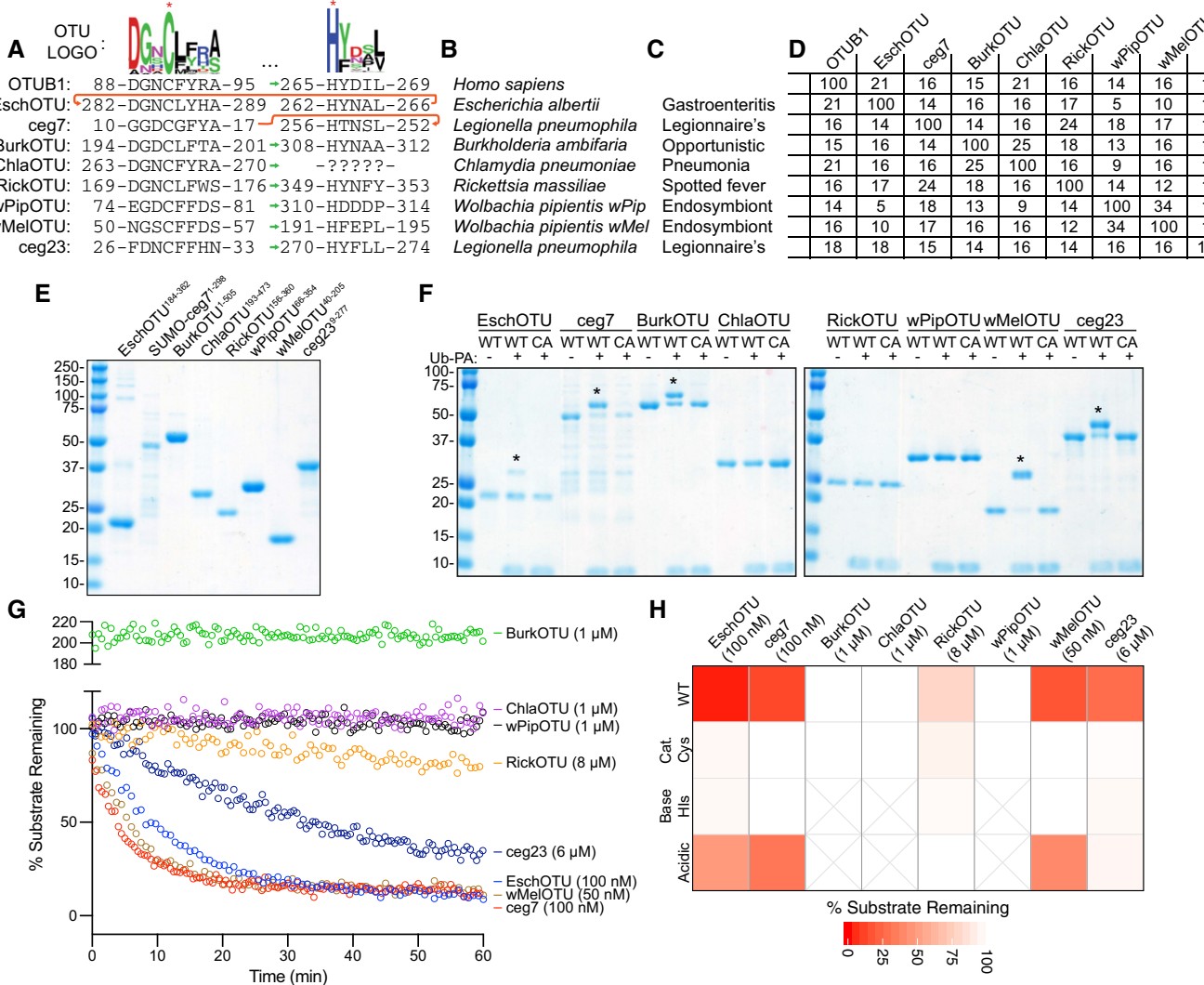

**Figure 1. Prediction and validation of OTU DUBs from bacteria.**

A   Pfam-generated sequence logo of the regions surrounding the OTU catalytic Cys and general base His (marked with asterisks). The conservation of these regions in the human OTUB1 and predicted bacterial OTUs are shown below, together with their relative order in the sequence topology indicated by the sequence position as well as green and red arrows for the typical and atypical arrangements, respectively.

B   Bacterial species to which the predicted OTUs belong.

C   Outcome of interactions between the highlighted bacterial species and their respective eukaryotic hosts.

D   Percent identity matrix calculated from a PSI-Coffee alignment (Notredame *et al*, 2000) of the predicted OTU domains. OTUB1 (80–271), EschOTU (184–362), ceg7 (1–298), BurkOTU (186–315), ChlaOTU (193–473), RickOTU (161–356), wPipOTU (66–354), wMelOTU (40–205), and ceg23 (9–277) were used to create the alignment.

E   Coomassie-stained SDS–PAGE gel showing purified protein from the predicted bacterial OTU constructs.

F   Ub-PA activity-based probe assay for wild-type (WT) and catalytic Cys-to-Ala mutants (CA). Strong, Cys-dependent reactivity is indicated with asterisks.

G   Ub-KG(TAMRA) cleavage assay monitored by fluorescence polarization at the indicated DUB concentrations. Note that BurkOTU displays an increase in fluorescence polarization, indicative of noncovalent binding.

H   Heatmap representation of DUB activity against the Ub-KG(TAMRA) substrate shown in (G), including the WT enzyme and Ala substitutions at the predicted catalytic Cys, general base His, or acidic position. Substrate remaining at the end of the assay is reported after correction against an initial reading from an equivalent assay performed with the catalytically inactive CA mutants.

Ub-Propargylamine (Ub-PA) activity-based probe that covalently reacts with a DUB's active site Cys, resulting in an 8.5 kDa shift in molecular weight on SDS–PAGE (Ekkebus *et al*, 2013). By this approach, EschOTU, ceg7, BurkOTU, wMelOTU, and ceg23 all showed robust reactivity with the Ub-PA probe that was abolished following mutation of the predicted active site Cys to Ala (Fig 1F). This assay validated some of our OTU predictions and our

identification of a catalytic Cys. To visualize genuine protease activity with improved sensitivity, we implemented a fluorescence polarization assay that detects the release of a C-terminal isopeptide-linked fluorescent peptide (Geurink *et al*, 2012). In addition to EschOTU, ceg7, wMelOTU, and ceg23, this assay could also detect DUB activity for RickOTU (albeit at high enzyme concentration) (Fig 1G). ChlaOTU, wPipOTU, and BurkOTU showed no activity

against this substrate, but BurkOTU did exhibit a dramatic increase in fluorescence polarization indicative of a strong interaction with the Ub substrate (Fig 1G). The observation of binding without cleavage could indicate that either BurkOTU has a high-affinity Ub binding site outside of the S1 site or the orientation of the catalytic site may be regulated through some other means. For those that demonstrated activity against the fluorescent Ub substrate, we additionally tested for dependence upon our predicted active site triad residues (catalytic Cys, general base His, and acidic). In all cases, mutation of the Cys or His residues to Ala abolished DUB activity (Figs 1H and EV1C). The acidic position is typically the second amino acid C-terminal to the general base His, and in similar manner to human OTUs, its mutation can result in complete, intermediate, or no loss of activity in the bacterial OTUs (Figs 1H and EV1C). Members in the A20 subfamily of human OTUs encode their acidic residue N-terminal to the catalytic Cys (Komander & Barford, 2008); we predicted a similarly positioned acidic residue in the ceg23 sequence (D21), and its mutation abolished DUB activity (Figs 1H and EV1C).

## Substrate specificities of bacterial OTU deubiquitinases

Across eukaryotic and viral examples, the OTU family has been shown to display a remarkable diversity in substrates specificities, both at the level of Ub/Ub-like specificity (e.g., Crimean Congo hemorrhagic fever virus vOTU dual Ub/ISG15 activity (Frias-Staheli *et al*, 2007; Akutsu *et al*, 2011; James *et al*, 2011)) and at the level of polyUb chain types [e.g., K11, K48, or M1 specificity (Mevissen *et al*, 2013)]. Therefore, we sought to assess our bacterial OTUs for both types of substrate specificity.

To measure Ub/Ub-like specificity, we used fluorescence polarization to measure activity toward Ub, ISG15, NEDD8, and SUMO1 in parallel (Figs 2A–C and EV2A). EschOTU, ceg7, RickOTU, and wMelOTU primarily targeted Ub under these conditions (Figs 2A and C, and EV2A). In addition to its activity toward the Ub substrate, ceg23 could also cleave the SUMO1 substrate (Figs 2C and EV2A). This particular combination of Ub/Ub-like proteolytic activities had previously only been observed in XopD from the plant pathogen *Xanthomonas campestris* (Pruneda *et al*, 2016). While BurkOTU did not demonstrate any cleavage of the Ub/Ub-like substrates, the increased signal indicative of an interaction with the Ub substrate was specific and was not observed with any of the Ub-like substrates (Fig 2B and C). ChlaOTU and wPipOTU showed no activity against any of the Ub/Ub-like substrates.

Specificity at the level of polyUb chain type was measured by constructing a panel of all eight canonical diUb linkages for use in gel-based cleavage assays (Mevissen *et al*, 2013; Michel *et al*, 2018). To better visualize any discrimination between chain types, enzyme concentration and incubation times were optimized such that at least one diUb species was nearly or completely cleaved by the end of the experiment (Figs 2D and E, and EV2B). Under no conditions were we able to observe activity for ChlaOTU or wPipOTU. All other bacterial OTUs (including BurkOTU) showed DUB activities with moderate discrimination between chain types (Fig 2F). Interestingly, EschOTU, ceg7, BurkOTU, RickOTU, wMelOTU, and ceg23 all shared a common basal preference for K6-, K11-, K48-, and K63-linked chains (Fig 2F), a combination not observed in any of the human OTU DUBs (Mevissen *et al*, 2013) but surprisingly similar to some viral OTUs (Dzimianski *et al*, 2019).

Among these chain types, there were some indications of further preference: EschOTU, ceg7, and RickOTU demonstrated a slight preference toward K48-linked chains, BurkOTU toward K11, wMelOTU toward K6, and ceg23 more strongly toward K63 linkages (Figs 2D–F and EV2B). Underneath these preferences were several lowly cleaved background activities, including K33-linked chains across all active examples and an additional activity toward M1-linked chains from ceg7. Notably, aside from reactivity with the Ub-PA probe, diUb cleavage offered the first robust measure of activity for BurkOTU and allowed for the confirmation of all three predicted active site triad residues by mutagenesis (Fig EV2C). The peculiar requirement of polyUb chains for BurkOTU activity is reminiscent of OTULIN (Keusekotten *et al*, 2013) and could indicate a mechanism by which binding to the S1' site drives substrate recognition and catalysis.

## Bacterial OTU deubiquitinases demonstrate novel modes of substrate recognition

To confirm that our validated bacterial DUBs are indeed members of the OTU family, we determined a crystal structure of wMelOTU to 1.5 Å resolution by molecular replacement with the core structure of yeast OTU1 (Messick *et al*, 2008; Figs 3A and EV3A, Table 1). The wMelOTU structure exhibits a pared-down canonical OTU domain architecture with a central β-sheet supported underneath by an α-helical subdomain, but although additional α-helical content typically sandwiches the β-sheet from above, there is very little additional support in the wMelOTU structure (Figs 3A and B, and EV3B). The core of the OTU fold that contains the active site (the central β-sheet and two most proximal supporting α-helices) closely resembles other OTU domains such as OTUB1 (Fig 3B, 1.6 Å RMSD) and vOTU (Fig EV3B, 1 Å RMSD), whereas the surrounding areas of structure are more divergent (Akutsu *et al*, 2011; Juang *et al*, 2012). Two regions of structure near the S1 substrate recognition site, encompassing 6 and 7 amino acids, respectively, are missing from the electron density (Figs 3A and EV3A). The structure confirms our prediction and mutagenesis of active site residues (Figs 1A and H, and 3A). However, the catalytic triad is misaligned (Fig 3A) as a result of the loop preceding the general base His (the so-called His-loop) occupying a descended conformation that would also occlude entry of the Ub C-terminus into the active site (Fig 3C). Thus, while the apo wMelOTU structure validates our prediction of an OTU fold, it raised new questions as to the mechanisms of substrate recognition.

Ub substrate recognition by wMelOTU was visualized by covalently trapping a wMelOTU-Ub complex and determining its crystal structure to 1.8 Å resolution (Figs 3D and EV3C, Table 1). As anticipated, the Ub C-terminus was found to be covalently linked to the wMelOTU catalytic Cys. The Ub-bound structure closely resembles the apo wMelOTU structure, with several key differences that provide insight into substrate recognition. Firstly, not only did Ub binding shift the His-loop up into position that opens entry into the active site, but in doing so it aligned the catalytic triad to facilitate nucleophilic attack (Fig 3E). The second major insight from the wMelOTU-Ub structure is the mode of Ub binding, which is very distinct from anything observed in previous OTU studies. The two regions of missing density in the apo wMelOTU structure are ordered in the Ub-bound complex as two β-hairpins

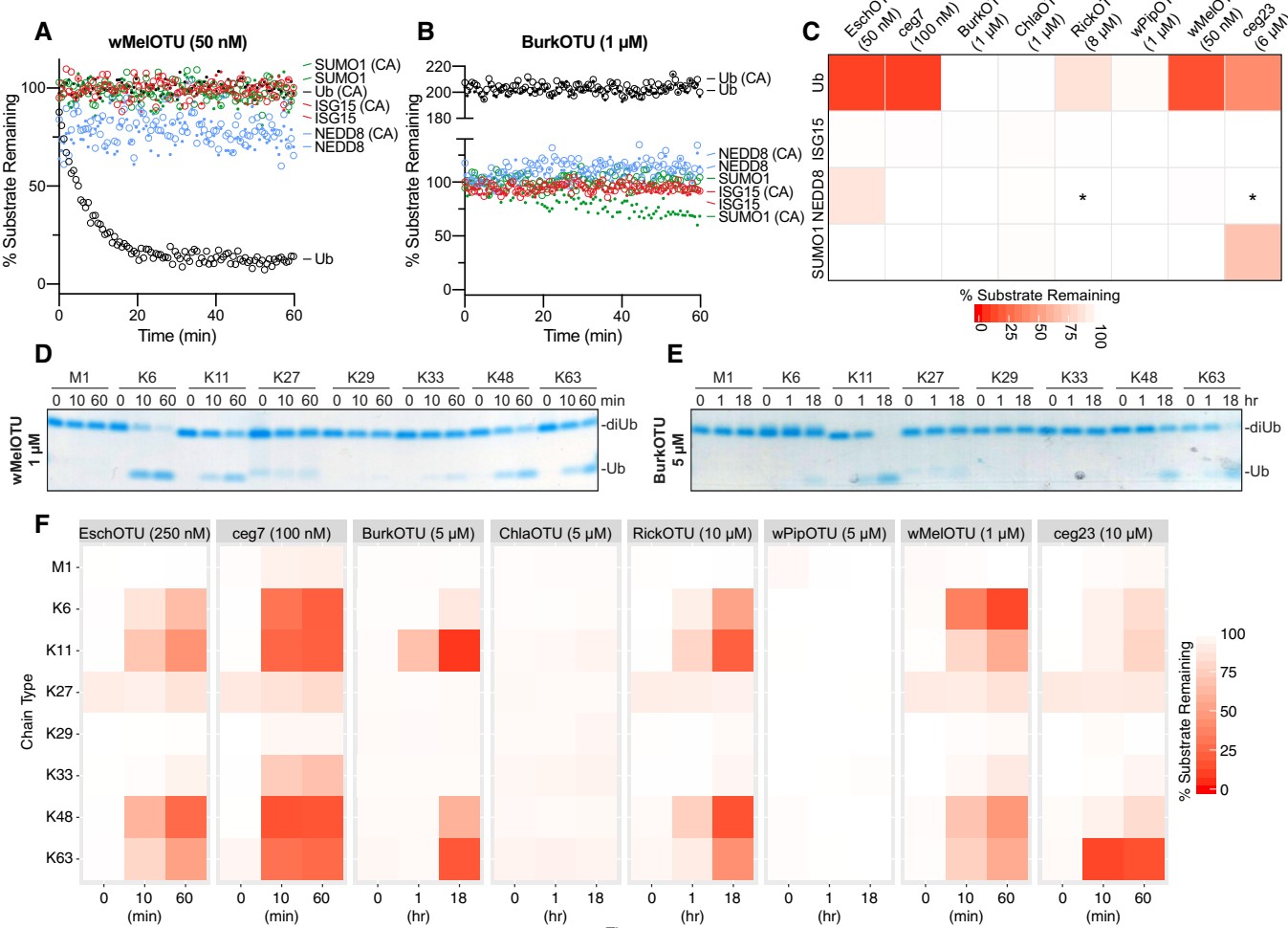

**Figure 2. Substrate specificity profiling of bacterial OTU DUBs.**

A  Ub/Ub-like specificity assay measuring activity of WT and inactive Cys-to-Ala wMelOTU toward the Ub-, ISG15-, NEDD8-, and SUMO1-KG(TAMRA) substrates.
B  Ub/Ub-like specificity assay measuring activity of WT and inactive Cys-to-Ala BurkOTU toward the Ub-, ISG15-, NEDD8-, and SUMO1-KG(TAMRA) substrates. Note that the rise in fluorescence polarization signal, indicative of a noncovalent interaction, is specific to the Ub substrate.
C  Heatmap representation of corrected OTU activities toward the Ub and Ub-like fluorescent substrates. In the reactions marked by an asterisk, an unusually high level of noise in fluorescence polarization signal was observed, likely a result of high OTU concentration.
D  Ub chain specificity assay measuring wMelOTU activity toward the eight diUb linkages. Reaction samples were quenched at the indicated timepoints, resolved by SDS–PAGE, and visualized by Coomassie staining.
E  Ub chain specificity assay measuring BurkOTU activity toward the eight diUb linkages. Reaction samples were quenched at the indicated timepoints, resolved by SDS–PAGE, and visualized by Coomassie staining.
F  Heatmap representation of WT bacterial OTU activities toward the eight diUb linkages at the indicated timepoints.

that wrap around the Ub, forming an embrace (Figs 3D and E, and EV3C). Together with additional interactions from a loop extending off the edge of the central β-sheet, wMelOTU forms a tripartite S1 site that becomes stabilized upon substrate binding (Fig 3D). Although this S1 site is on a similar surface of the OTU domain, the distinctive recognition elements (to be discussed in a broader context below) position the bound Ub moiety in a drastically different orientation that is 107° or 167° rotated from the vOTU-Ub or OTUB1:Ub structures, respectively (Fig EV3D; Akutsu *et al*, 2011; Juang *et al*, 2012).

The primary and secondary contacts to Ub form the bulk of the interaction and arise from the two stabilized β-hairpins (Fig 3D and

F). The primary hairpin extends from the central β-sheet and forms hydrophobic interactions with the I44 hydrophobic patch of Ub. L154, L156, and V149 of wMelOTU are buried in hydrophobic interactions with Ub L8, I44, H68, and V70 (Fig 3F). The secondary β-hairpin replaces what is typically a helical arm in other OTUs and contacts the Ub I36 hydrophobic patch with H99 (Fig 3F). Q147 from the primary β-hairpin of wMelOTU forms a hydrogen bond to the carbonyl backbone of Ub L71, but also to the side chain of N101 from the secondary β-hairpin as if to lock the embrace (Fig 3F). Mutations at any of the Ub-contacting wMelOTU residues negatively impact DUB activity (Fig 3G). Moving into the active site, R72 of the Ub C-terminus is coordinated by hydrogen bonds to the backbone of

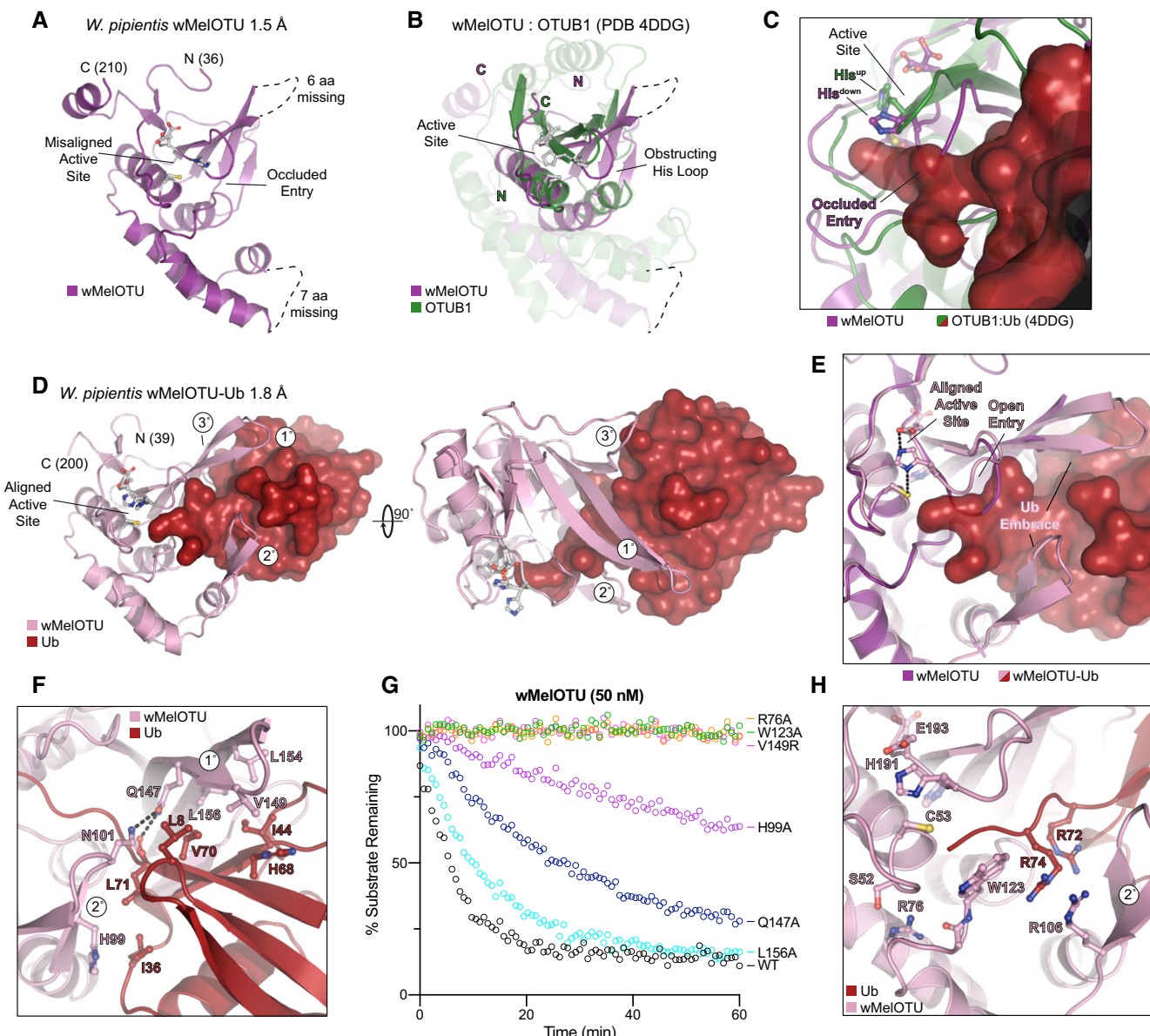

**Figure 3.  wMelOTU structure reveals novel Ub embrace mechanism.**

A   Cartoon representation of the 1.5 Å *Wolbachia pipientis* wMelOTU crystal structure with labeled termini, missing regions, and features of the active site.

B   Structural alignment of the core OTU folds (central β-sheet and two supporting α-helices) from human OTUB1 (green, PDB 4DDG) and wMelOTU (purple). Surrounding regions are less well conserved and shown as semi-transparent.

C   Enlarged region of the OTUB1:Ub structure (PDB 4DDG) showing entry of the Ub C-terminus (red) into the OTUB1 active site (green). The wMelOTU structure (purple) is overlaid to highlight the structural conflict between the downward position of the His-loop and the Ub C-terminus.

D   1.8 Å crystal structure of the covalent wMelOTU-Ub complex. wMelOTU (cartoon, pink) is linked to the Ub (surface, red) C-terminus through its active site. Primary, secondary, and tertiary regions of the Ub-binding S1 site are indicated.

E   Structural overlay of the apo (violet) and Ub-bound (pink) wMelOTU structures highlighting the repositioning of the His-loop to accommodate entry of the Ub C-terminus, as well as ordering of two regions in the S1 site that form an embrace around Ub.

F   Detailed view of the primary and secondary interfaces between wMelOTU (pink) and Ub (red) observed in the wMelOTU-Ub structure. wMelOTU and Ub residues participating in the interface are shown with ball and stick representation.

G   Ub-KG(TAMRA) cleavage assay monitoring the effects of structure-guided wMelOTU mutations. These data were collected in parallel with those presented in Fig 1G, and the WT dataset is shown again for reference.

H   Detailed view of the wMelOTU (pink) active site region and its coordination of the Ub C-terminus (red). Residues that coordinate Ub or stabilize the active site are shown with ball and stick representation.

**Table 1. Data collection and refinement statistics.**

| | wMelOTU | wMelOTU-Ub | EschOTU-Ub |
|---|---|---|---|
| Data collection | | | |
| Space group | $P\,2_1\,2_1\,2_1$ | $C\,1\,2\,1$ | $P\,4_1\,2\,2$ |
| Cell dimensions | | | |
| $a, b, c$ (Å) | 52.54, 56.64, 63.96 | 136.93, 78.20, 280.43 | 67.34, 67.34, 144.43 |
| α, β, γ (°) | 90, 90, 90 | 90, 91.57, 90 | 90, 90, 90 |
| Resolution (Å) | 33.00–1.47 (1.52–1.47) | 27.29–1.82 (1.89-1.82) | 67.34–2.10 (2.18–2.10) |
| $R_{merge}$ | 0.049 (0.678) | 0.139 (0.867) | 0.032 (0.884) |
| $I/\sigma I$ | 15.4 (2.7) | 9.4 (2.8) | 14.5 (1.5) |
| Completeness (%) | 99.52 (99.76) | 92.21 (94.66) | 99.5 (99.6) |
| Redundancy | 4.3 (4.1) | 7.8 (7.7) | 4.2 (4.4) |
| Refinement | | | |
| Resolution (Å) | 33.00–1.47 | 27.29–1.82 | 67.34–2.10 |
| No. unique reflections/test set | 33,002/3,265 | 244,560/24,941 | 20,025/1,972 |
| $R_{work}/R_{free}$ | 0.162/0.189 | 0.167/0.208 | 0.218/0.256 |
| No. atoms | | | |
| Protein | 1,309 | 22775 | 1,961 |
| Ligand/ion | 4 | 192 | 12 |
| Water | 185 | 3,647 | 63 |
| B-factors | | | |
| Protein | 24.8 | 22.7 | 69.2 |
| Ligand/ion | 58.1 | 26.6 | 76.4 |
| Water | 42.0 | 34.5 | 67.7 |
| R.m.s. deviations | | | |
| Bond lengths (Å) | 0.014 | 0.009 | 0.015 |
| Bond angles (°) | 1.36 | 0.90 | 1.26 |

Values in parentheses are for highest resolution shell.

the secondary β-hairpin, which also positions wMelOTU R106 to stack with Ub R74 (Fig 3H). Proximal to the active site, wMelOTU displays several conserved features of the OTU fold. Firstly, the GlyGly motif is held in place by wMelOTU with a conserved aromatic residue, W123 (Fig 3H). Secondly, a conserved basic residue, R76, supports both the loop containing W123 and the loop preceding the catalytic Cys (the so-called Cys-loop) that forms the oxyanion hole (Fig 3H). Mutation at either of these conserved positions abrogates DUB activity (Fig 3G). In sum, though many features of the wMelOTU fold and active site arrangement are reminiscent of eukaryotic and viral OTUs, Ub recognition within the S1 site itself is distinct from previously studied examples.

**An alternate topological arrangement of the OTU fold**

Intrigued by our prediction of an alternate threading through the OTU fold of EschOTU (Fig 1A), we sought to validate its sequence

topology by determining a structure. A crystal structure of a covalent EschOTU-Ub complex was determined to 2.1 Å resolution by molecular replacement with Ub and a sieved model of the OTU domain generated using MUSTANG (Konagurthu *et al*, 2010; Figs 4A and D, and EV4A, Table 1). The structure confirms our predicted and tested active site residues (Figs 1A and H, and 4A) as well as the overall OTU domain architecture. Like wMelOTU, the EschOTU OTU domain is a pared-down version that aligns well with OTUB1 and vOTU through the central β-sheet and supporting α-helices (0.6 and 0.5 Å RMSD, respectively) (Fig 4B and C), but lacks α-helices above the sheet that would form the canonical sandwich structure. Perhaps the most remarkable insight, which will be discussed in a broader context below, is the permutation of the N- and C-termini that leads to altered threading through the OTU fold. While the termini are typically in close proximity above the central β-sheet in all other known OTU folds, EschOTU threads a loop at this position and the termini are instead located in the supporting helical region beneath the sheet, near the helical arm of the S1 site (Fig 4A–C). Another interesting feature observed in the crystal lattice is how an N-terminal region (aa 184–192) from a symmetry-related EschOTU molecule adds an additional strand onto the edge of the central β-sheet (Figs 4A, and EV4B and C). Although this strand aligns well with structurally related strands in OTUB1 and vOTU (Fig 4B and C), its removal has no effect on DUB activity (Fig EV4D), and thus, we believe its position was a result of crystallization.

The Ub-binding S1 site is comprised almost entirely of a primary interaction between a helical arm region and the I44 hydrophobic patch of Ub, and makes very few contacts through what is normally a secondary interaction site in other OTUs (Fig 4D–F). The bound Ub is held in an orientation distinct from the vOTU-Ub structure (95° rotation, Fig 4D and F) but very similar to that observed in OTUB1 and other closely related OTUs (21° rotation, Fig 4D and E). At the primary site of interaction, the EschOTU helical arm residues C338 and I341, as well as nearby L224, all contact the I44 hydrophobic patch of the bound Ub (Fig 4G), and mutation of these positions results in diminished DUB activity (Fig 4H). A small secondary interaction site is formed between L241 in the edge strand of the EschOTU central β-sheet and the Ub I36 hydrophobic patch (Fig 4G). Although this interaction surface is smaller, it likely plays an important role in coordinating Ub L71 and L73 as the C-terminus enters the active site, and accordingly, mutation of L241 also decreases DUB activity (Fig 4H). In a similar theme to wMelOTU and other OTU examples, structural elements close to the active site are much more conserved. R74 in the Ub C-terminus is coordinated by EschOTU E343, the GlyGly motif is secured by W214, and the Cys-loop is stabilized by the conserved basic residue K318 (Fig 4I). Mutation at any of these EschOTU positions diminishes or abrogates DUB activity (Fig 4H). Altogether, unlike wMelOTU, the S1 site of EschOTU more closely resembles canonical OTUs with a familiar helical arm. The sequence topology of the EschOTU fold, however, is distinct from all other OTU structures and suggests an interesting evolutionary history that is discussed in more detail below.

**A cross-kingdom analysis of the OTU fold**

Our diverse list of confirmed bacterial OTU DUBs and representative crystal structures afforded the first opportunity for a cross-kingdom

analysis of the OTU fold across eukaryotes and prokaryotes, as well as viruses. Because of the significantly altered topology we observed in the EschOTU structure (Figs 1A and 5A), we focused our first analysis on the threading of the OTU domain. Human OTUB1 and vOTU represent the most typical arrangement, wherein the N- and C-termini of the OTU domain are positioned near each other in the α-helical region above the central β-sheet (Fig 5B, open gray arrow), and the catalytic triad is threaded in the C…H-Ω-D/N/E arrangement (where Ω represents a large aromatic residue) (Figs 1A and 5B). EschOTU, however, encodes a reversed H-Ω-N…C arrangement of the catalytic triad as a result of a sequence permutation that closes the traditional N- and C-termini into a loop (Fig 5A and B,

compare open and closed gray arrows) and opens new termini near the helical arm region (Fig 5A and B, compare open and closed black arrows). A third arrangement of the catalytic triad is represented by members of the A20 subfamily of OTUs (Komander & Barford, 2008; Mevissen *et al*, 2016; Fig 5C). Instead of encoding the acidic triad residue on the same β-strand as the general base His, A20-family OTUs encode this residue before the catalytic Cys and position it directly above the β-sheet in tertiary structure (Komander & Barford, 2008; Fig 5C).

Our structure of wMelOTU shows that its sequence topology matches the most typical OTU arrangement seen in OTUB1 and vOTU (Figs 3A and 5B), and we would predict BurkOTU, RickOTU,

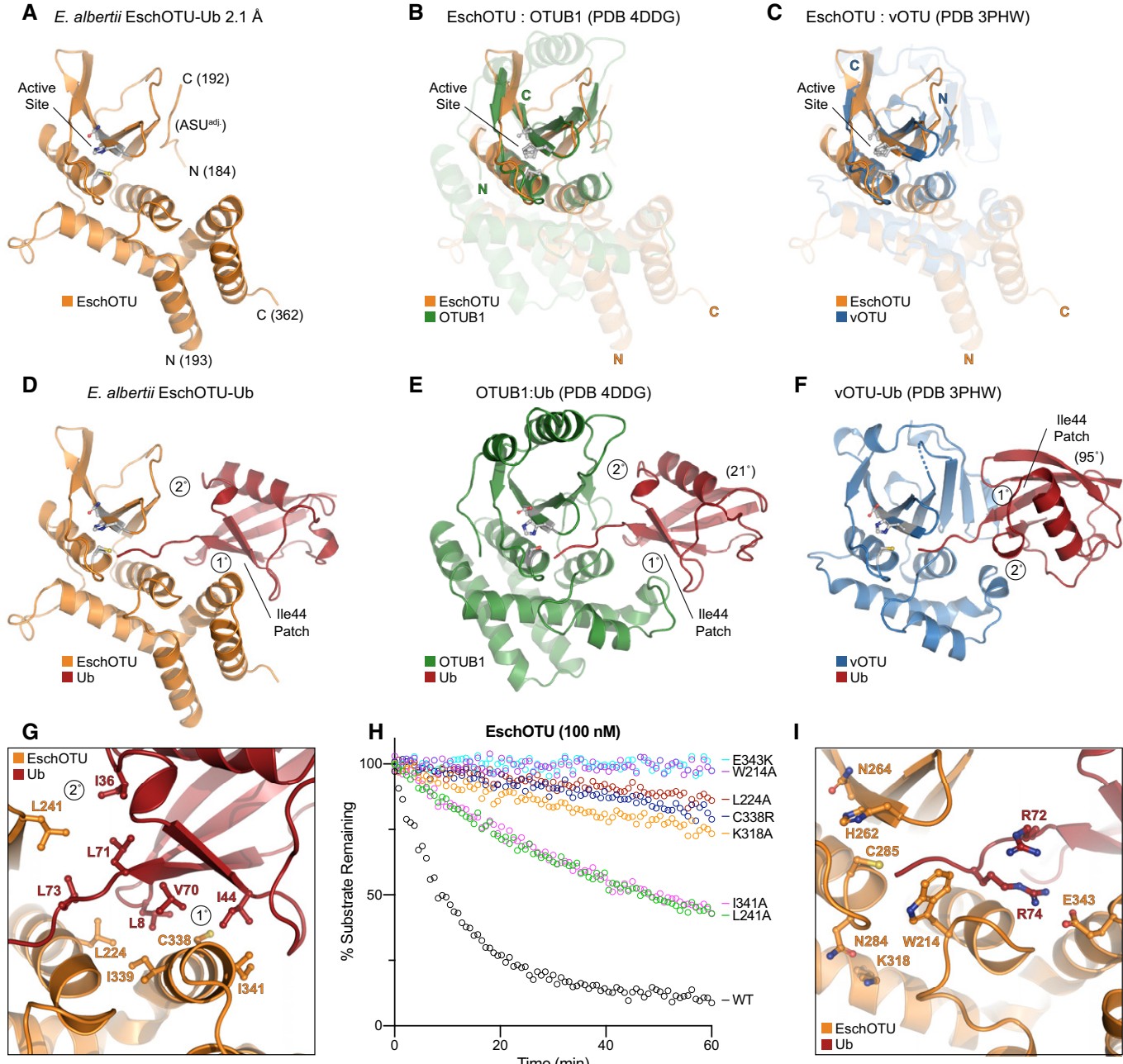

**Figure 4.**

**Figure 4.  EschOTU structure shows altered sequence topology.**

A   Cartoon representation of the 2.1 Å *Escherichia albertii* EschOTU-Ub crystal structure with labeled termini and active site. Ub is hidden for this initial view of the OTU fold.

B   Structural alignment of the core OTU folds (central β-sheet and two supporting α-helices) from human OTUB1 (green, PDB 4DDG) and EschOTU (orange). Surrounding regions are less well conserved and shown as semi-transparent.

C   Structural alignment of the core OTU folds (central β-sheet and two supporting α-helices) from CCHFV vOTU (blue, PDB 3PHW) and EschOTU (orange). Surrounding regions are less well conserved and shown as semi-transparent.

D   Full view of EschOTU (orange) covalently bound to Ub (red) in the S1 site. Primary and secondary interactions with Ub are labeled, as well as the Ub Ile44 hydrophobic patch.

E   An aligned view as in (D), showing S1 site interactions between human OTUB1 (green) and Ub (red) (PDB 4DDG). Ub is rotated 21° relative to the EschOTU-Ub structure, but maintains similar primary and secondary contacts.

F   An aligned view as in (D), showing S1 site interactions between CCHFV vOTU (blue) and Ub (red) (PDB 3PHW). Ub is rotated by 95° relative to the EschOTU-Ub structure and displays swapped primary and secondary contacts.

G   Detailed view of the primary and secondary interfaces observed in the EschOTU-Ub structure. EschOTU (orange) and Ub (red) residues participating in the interface are shown with ball and stick representation.

H   Ub-KG(TAMRA) cleavage assay monitoring the effects of structure-guided EschOTU mutations. These data were collected in parallel with those presented in Fig 1G, and the WT dataset is shown again for clarity.

I   Detailed view of the EschOTU (orange) active site region and its coordination of the Ub C-terminus (red). Residues that coordinate Ub or stabilize the active site are shown with ball and stick representation.

and wPipOTU to be similar as well (Fig 1A). Our alignment and mutagenesis data would suggest that *Legionella* ceg23 is most similar to the A20 sequence topology and positions the acidic D21 residue above the remaining C29 and H270 triad residues (Figs 1H and 5C, and EV1C). A recent crystal structure of ceg23 confirms our prediction of the active site topology (Ma *et al*, 2020). Based on our secondary structure and catalytic motif analyses, we would predict that *Legionella* ceg7 adopts yet another topology such that the β-strand encoding the general base His is threaded in the opposite direction (Fig 1A); testing this arrangement, however, awaits structure determination.

To test whether a simple permutation of the OTU sequence was still permissive to protein folding and DUB activity, we rearranged the sequence of CCHFV vOTU to match the altered topology observed in EschOTU (compare Fig 5A and D). By closing a loop (Fig 5D, gray arrow) and opening new N- and C-termini (Fig 5D, black arrow), we were able to generate a permutated vOTU variant (vOTU$^P$) that mimicked the EschOTU sequence topology. Despite the altered threading, vOTU$^P$ was still folded and could be modified by the Ub-PA activity-based probe (Fig 5E). vOTU$^P$ also demonstrated cleavage of the Ub-KG(TAMRA) substrate, though to a lesser degree than the wild-type topology (Fig 5F). Thus, the OTU fold is amenable to permutation as well as to the repositioning of catalytic residues, which may make future sequence analysis of this and other highly divergent examples of the OTU fold more difficult.

**A framework for understanding the S1 site of OTU domains**

Because we were able to determine structures of wMelOTU and EschOTU with substrate Ub bound, we could also use this new information to better describe elements of the S1 site that are either common or distinctive across eukaryotic, bacterial, and viral OTUs. Owing to its basic role in establishing DUB activity in OTUs, one would expect the S1 site to be somewhat conserved (as opposed to other sites, such as S1', that further discriminate the type of Ub substrate); however, we note a remarkable variability in the structural elements used to contact Ub. Surrounding a commonly positioned helix (constant region, CR), we could define three regions of variability (variable regions, VR) that together form the S1 site (Fig 6A).

The first region, VR1, is often the primary site of interaction and is typically referred to as the helical arm (henceforth we propose to coin this region as simply "arm"). Adaptation of the VR1 arm region can be observed as either a short α-helix (e.g., in the Otubain or OTUD subfamilies), an extended α-helical region (e.g., in EschOTU, ceg23, or the A20 subfamily), or even a β-hairpin (e.g., in wMelOTU) (Fig 6A and B). As we noted with other VRs, different OTU VR1s can be used to contact different interaction surfaces of the Ub substrate, including the I44 or I36 hydrophobic patches (Fig EV5A). We defined VR2 as the edge of the central β-sheet (Fig 6A), which in addition to the common configuration of β-strands (e.g., in the Otubain or OTUD subfamilies), can be extended by additional β-strands (e.g., in vOTU or wMelOTU), or contracted (e.g., in EschOTU) (Fig 6C). Additionally, the arterivirus PLP2 encodes an inserted zinc finger at VR2 that forms the basis for its interaction with Ub (Fig 6C; van Kasteren *et al*, 2013). This VR2 edge can be used to contact Ub surfaces such as the I44 or I36 hydrophobic patches, or in the case of wMelOTU the D58 acidic patch (Fig EV5B). The final variable region identified in our analysis, VR3, is a β-turn in the central β-sheet (Fig 6A) that can be short (e.g., in the OTUD subfamily or EschOTU), extended but unstructured (e.g., in the Otubain subfamily or vOTU), or extended to form a β-hairpin (e.g., in the A20 subfamily or wMelOTU) (Fig 6D). This VR3 region has been observed either to be unutilized for Ub recognition or to contact the I44 or I36 hydrophobic patches (Fig EV5C).

Together, by analyzing the S1 substrate recognition sites of eukaryotic, bacterial, and viral OTUs, we have identified surprising diversity confined to common regions of the fold. These variable regions can be adapted in a number of ways and can accommodate diverse orientations of substrate binding. Through cataloging the multiple adaptations of the S1 site, we have established a framework for future OTU domain analysis.

# Discussion

Our prediction and validation of OTU DUBs across a range of evolutionarily distinct bacteria has highlighted a number of distinguishing features in the enzyme fold and mechanism, and in addition

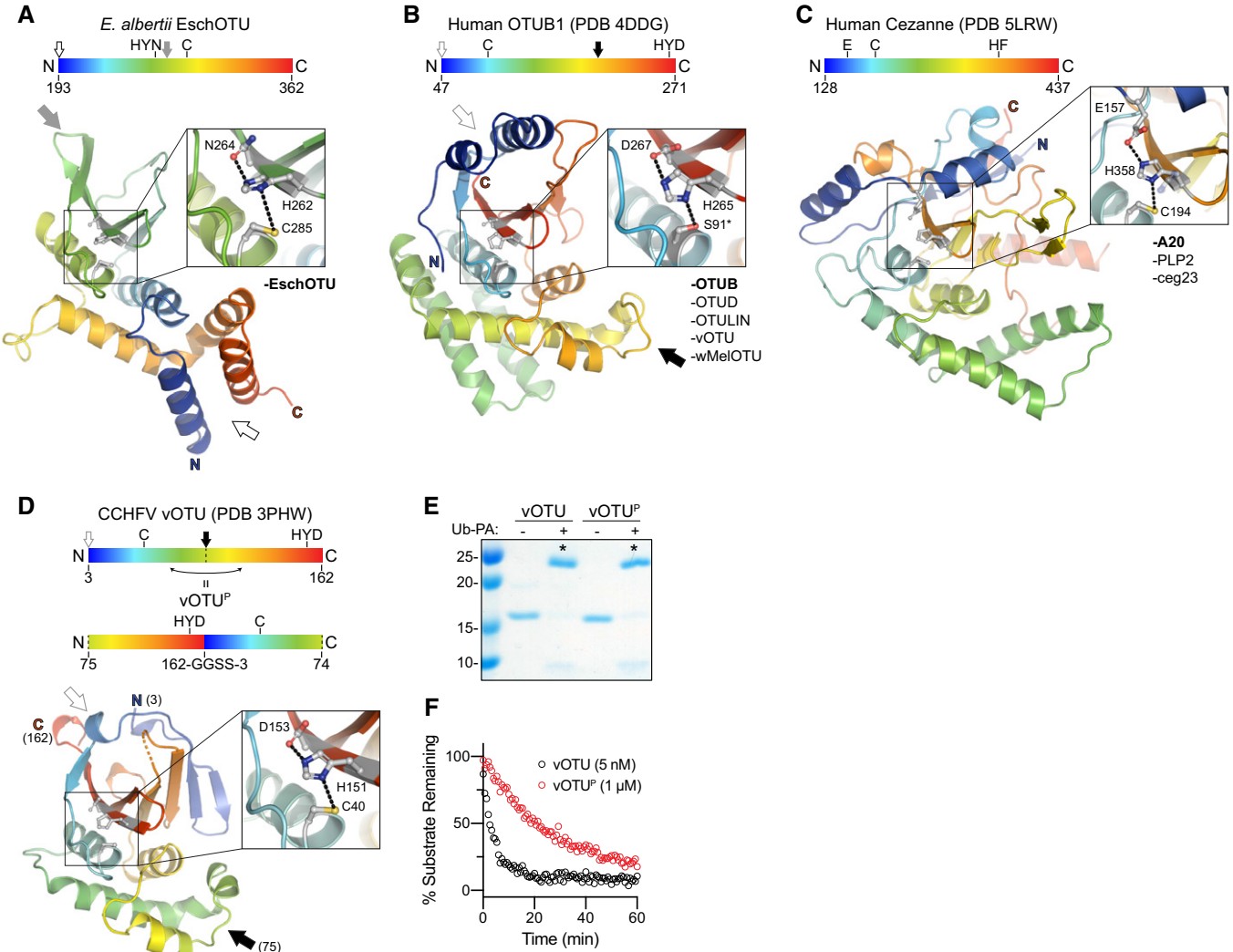

**Figure 5. Cross-kingdom structural analysis of the OTU fold.**

A  Cartoon representation of the EschOTU crystal structure colored in a rainbow gradient from N- to C-terminus. The catalytic triad residues are marked on both the structure and the linear color gradient above, showing their positions with respect to each other and the overall OTU sequence. The black and gray arrows relate how the EschOTU fold is permutated with respect to other OTUs. The black open arrow marks the open N- and C-termini, while the closed gray arrow marks a closed loop. OTU subfamilies that follow this overall sequence topology are listed in the lower right. This arrangement is only observed in EschOTU.

B  As in (A), for the human OTUB1 structure (PDB 4DDG). The closed black arrow marks a closed loop, while the open gray arrow marks the open N- and C-termini. This arrangement is representative of the human Otubain, OTUD, and OTULIN subfamilies, as well as vOTUs.

C  As in (A), for the human Cezanne structure (PDB 5LRW). This arrangement is representative of the human A20 subfamily, viral PLP2, and *Legionella* ceg23.

D  As in (A), for the CCHFV vOTU structure (PDB 3PHW). A schematic for the permutated vOTU$^P$ variant is shown to illustrate how it relates to the native sequence topology.

E  Ub-PA activity-based probe assay for WT vOTU and sequence-permutated vOTU$^P$. Strong reactivity is indicated with asterisks.

F  Ub-KG(TAMRA) cleavage assay monitored by fluorescence polarization for WT vOTU and sequence-permutated vOTU$^P$.

suggests that the OTU fold is an evolutionarily common and adaptable fold among eukaryotes, viruses, and bacteria. Given the low sequence similarity among our selected bacterial OTU domains, the similarities observed in Ub/Ub-like and polyUb chain specificities were surprising. All the active OTUs we identified targeted Ub preferentially over the Ub-like modifiers ISG15, NEDD8, or SUMO1. In addition to its DUB activity, in our assays *Legionella* ceg23 also cleaved the SUMO1 substrate, which could indicate a role for

SUMO1 signaling in restricting *Legionella* growth. The overall preference toward Ub signals is reflective of the specificity observed in human OTUs (Mevissen *et al*, 2013), whereas viral OTUs have evolved to target both Ub and antiviral ISG15 signaling (Frias-Staheli *et al*, 2007). At the level of chain specificity, we noted a common, underlying preference for K6-, K11-, K48-, and K63-linked chains and only slight biases toward particular chain types in certain examples. A lack of chain specificity is not uncommon

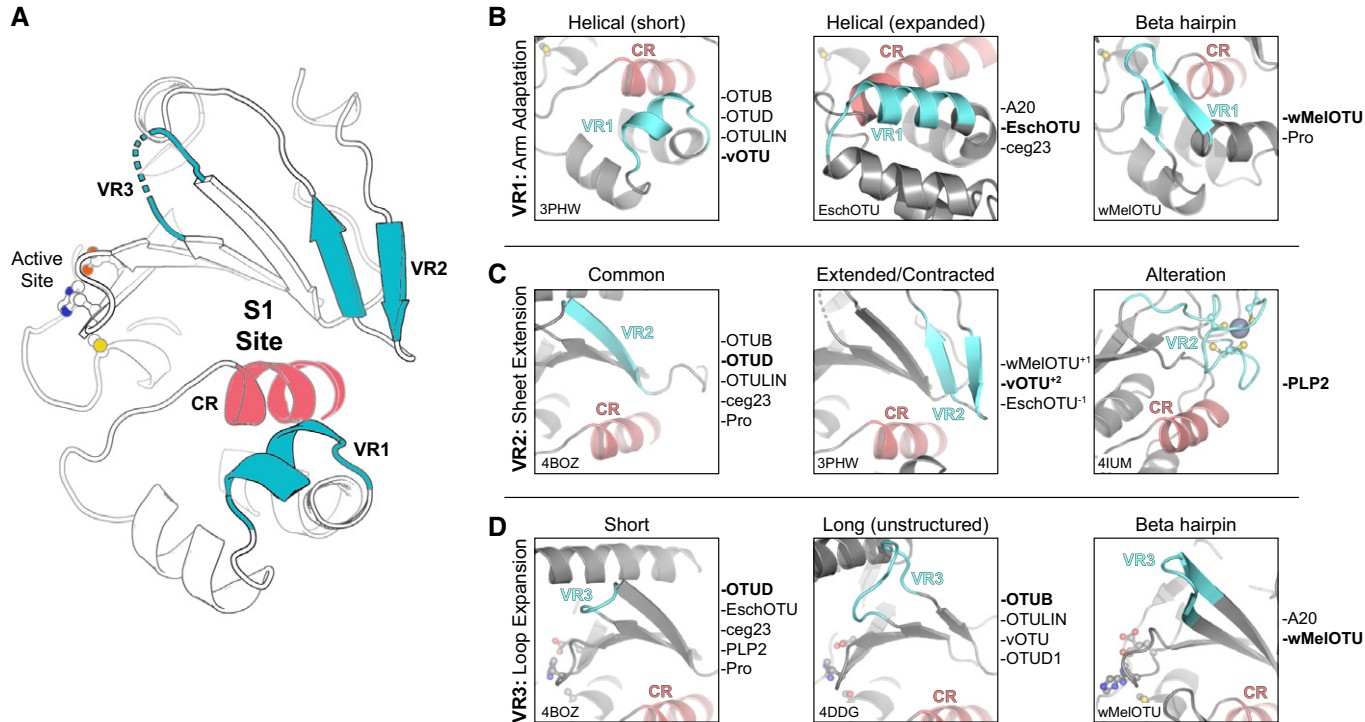

**Figure 6. A framework for understanding the S1 site of OTU domains.**

A   Cartoon representation of the OTU fold (vOTU, PDB 3PHW), with the active site and S1 site indicated. The S1 site is composed of a common region (CR, red) surrounded by three variable regions (VR, blue) that are responsible for Ub binding.

B   Comparison of structural adaptations in the VR1 arm region of the S1 site. VR1 has been observed to contribute to Ub binding as either a short α-helical segment (left), an extended α-helical region (center), or a β-hairpin (right). Examples of OTUs that follow each arrangement are provided to the right.

C   Comparison of structural adaptations in the VR2 central β-sheet edge of the S1 site. VR2 has been observed to contribute to Ub binding in its most common arrangement (left), with additional or fewer β-strands (center), or altered with additional substructure (right). Examples of OTUs that follow each arrangement are provided to the right.

D   Comparison of structural adaptations in the VR3 loop extending from the central β-sheet. This VR3 loop has been observed as short and not utilized in Ub binding (left), expanded and participating in unstructured interactions with Ub (center), or expanded with a β-hairpin motif that binds Ub (right). Examples of OTUs that follow each arrangement are provided to the right.

among OTUs (Mevissen *et al*, 2013; Dzimianski *et al*, 2019), but contrasts the co-evolved preferences for K63-linked chains observed among bacterial CE clan DUBs (Pruneda *et al*, 2016). Some human DUBs require accessory domains, proteins, or post-translational modifications to acquire their chain specificity (Mevissen & Komander, 2017). Human OTUD5, for example, demonstrates phosphorylation-dependent activity (Huang *et al*, 2012). It is possible that the bacterial OTUs leverage some unknown host cofactors or modifications to fine-tune or, in the cases of ChlaOTU and wPipOTU, activate their DUB functions. Several other families of bacterial effectors require binding to host cofactors, including CE clan acetyltransferases (Mittal *et al*, 2010), the *Shigella flexneri* kinase OspG (Pruneda *et al*, 2014), and the *Pseudomonas aeruginosa* phospholipase ExoU (Anderson *et al*, 2011). Alternatively, polyUb chain specificity may not be selected for, as appears to be the case for viral OTUs (Dzimianski *et al*, 2019), or the slight chain preferences we observed may reflect early signs of evolving specificities. Among the bacterial DUBs studied thus far, the *in vitro* substrate specificities measured against polyUb chains typically agree with their cellular roles in restricting chain-specific Ub signaling in, for example, the autophagic or inflammatory responses (Mesquita *et al*, 2012; Wan

*et al*, 2019). It is also possible, however, that some bacterial DUBs (including the OTUs we describe) may have instead evolved specificity for the ubiquitinated substrate, as is the case for some human DUBs (e.g., Morgan *et al*, 2016).

Similar to bacterial CE clan DUBs (Pruneda *et al*, 2016), we observe a remarkable diversity in the evolution of the S1 substrate-binding site among our bacterial OTUs (Fig 6A–D). wMelOTU in particular uses a disorder-to-order transition to embrace the Ub moiety through both of its commonly used hydrophobic patches. Whether these S1 site features have evolved to suit each organism's particular host–microbe interactions or reflect convergent evolution of DUB activity from a common protease scaffold remains an open question. The diversity in the S1 site among bacterial OTUs is in stark contrast to nairovirus OTUs, however, which appear to have made comparatively minor adjustments to a common template (Dzimianski *et al*, 2019). Regardless, through comparison of OTU: Ub recognition across eukaryotes, viruses, and bacteria we have identified three regions of sequence and structural variability that together form the substrate-binding S1 site (Fig 6A). The arm (VR1), β-sheet edge (VR2), and extended β-turn (VR3) can recognize any number of common interaction surfaces on Ub (Fig EV5A–C).

Interactions within the immediate vicinity of the OTU active site appear to be the only universal requirements for Ub recognition at the S1 site (Figs 3H and 4I). Alterations in the S1 site may be one method of tuning the level of DUB activity through evolution. As is the case with human OTUs (Mevissen *et al*, 2013), we observe a wide range of activities across the bacterial OTUs tested (Figs 1G and EV2B).

Classically, evolutionarily distinct clans of cysteine proteases have been classified by differences in tertiary structure as well as the linear topological arrangement of catalytic residues (Barrett & Rawlings, 1996). In this way, even though the CA clan (which encompasses all known human cysteine-dependent DUBs) and CE clan (including all human ULPs) are structurally related, they are classified separately due in large part to the threading of the active site: CA proteases encode the catalytic Cys before the general base His, whereas CE proteases are the reverse (Fig EV6A). With its permutated OTU fold (Figs 1A and 5A), classifying EschOTU into a protease clan is less straightforward (Figs 4B and EV6B). It has been proposed in the MEROPS (Rawlings *et al*, 2018) and SCOP (Fox *et al*, 2014) databases that CA and CE proteases share a common ancestor and have since undergone circular permutation. Since that event occurred, other changes have arisen that further distinguish the clans, namely the position of the acidic component of the catalytic triad. Aside from the A20 subfamily members which encode their acidic residue N-terminal to the catalytic Cys, all OTUs follow a common trend of the acidic residue being positioned two amino acids C-terminal of the general base His on the same β-strand. This is distinct from CE proteases that encode their acidic residue on a neighboring strand. Although EschOTU does encode an acidic residue (D278) at a position structurally analogous to acidic residues in CE clan catalytic triads, it is spatially too far (> 6 Å) to support the general base H262. EschOTU N264, on the other hand, is in the correct position for a catalytic triad (Fig 5A), and mutagenesis data confirm its role in DUB activity (Figs 1H and EV1C). Thus, despite its reversed sequence topology, we propose that EschOTU is more closely related to the OTU family of the CA protease clan and may either represent an evolutionary intermediate between the CA and CE clans or reflect an additional circular permutation of the fold.

Among our validated bacterial OTUs, we noted a common threshold of ~ 15% sequence identity to the human OTUB1 sequence (Fig 1D). This likely reflects a hard cutoff of our approach to prediction, as opposed to the true minimal conservation of the OTU domain itself. Considering both the potential for diversity of VRs in the S1 site and altered sequence topology, bioinformatic efforts to identify additional, possibly more divergent OTUs will be challenging. It is possible that through additional cross-kingdom analysis of the OTU fold, underlying structural and functional elements will be revealed that can assist with the further prediction of even more distantly related OTU domains in diverse bacteria.

# Materials and Methods

## Bacterial OTU prediction

To search for sequence-divergent OTU domains in bacteria, a multiple sequence alignment of all established OTU DUBs from eukaryotic and viral origin was generated using the L-INS-I algorithm of the MAFFT package (Katoh *et al*, 2002). From this alignment, a generalized sequence profile was constructed, scaled, and subjected to iterative refinement using the PFTOOLS package (Bucher *et al*, 1996). The final profile was run against a current version of the UniProt database. Matches to bacterial sequences with *P*-values < 0.01 were submitted to the Phyre2 web portal for secondary structure prediction and domain recognition (Kelley *et al*, 2015). Results were manually inspected for conservation of the active site Cys and His motifs described in Pfam (Entry PF02338) within α-helical and β-strand secondary structure, respectively. Type III and type IV secretion signals were predicted using the pEFFECT (Goldberg *et al*, 2016) and S4TE 2.0 (Noroy *et al*, 2019) web servers, respectively, with default parameters. pEFFECT predictions were made using the support vector machine approach.

## Construct design and cloning

With the exception of ceg23, which was cloned from *L. pneumophila* subsp. Pneumophila (strain Philadelphia) genomic DNA, all selected bacterial OTU genes were codon-optimized for *E. coli* expression and synthesized (GeneArt). Where possible, constructs were designed to include a minimal OTU domain based on active site and secondary structure analysis using Phyre2. In the case of BurkOTU, a longer construct that included a putative N-terminal domain was required to obtain soluble protein expression. EschOTU (184–362), ceg7 (1–298), and RickOTU (156–360) were cloned into the pOPIN-S *E. coli* expression vector (Berrow *et al*, 2007) that encodes an N-terminal His-SUMO tag. BurkOTU (1–505), ChlaOTU (193–473), wPipOTU (66–354), wMelOTU (40–205 or 1–215), and ceg23 (9–277) were cloned into the pOPIN-B *E. coli* expression vector (Berrow *et al*, 2007) that encodes an N-terminal, 3C protease cleavable His tag. EschOTU (184–362) and EschOTU (195–362) were additionally cloned into the pOPIN-B vector for comparison of activities. CCHFV vOTU (3–162) was cloned into pOPIN-B. The permutated vOTU$^{P}$ was generated by moving residues 75–162 upstream of residues 3–74, with a GlyGlySerSer linker encoded between the two.

## Protein expression and purification

All bacterial OTUs were expressed and purified with a similar approach. Transformed Rosetta 2 (DE3) *E. coli* were grown in LB at 37°C to an optical density (600 nm) of 0.6–0.8, at which point the culture was cooled to 18°C and induced with 0.2 mM IPTG for 16 h. Bacteria were harvested, resuspended in lysis buffer (25 mM Tris, 200 mM NaCl, 2 mM β-mercaptoethanol, pH 8.0), and subjected to one freeze–thaw cycle. The cells were then incubated on ice with lysozyme, DNase, and protease inhibitor cocktail (SIGMAFAST, Sigma-Aldrich) for 30 min, followed by lysis with sonication. The clarified lysates were applied to cobalt affinity resin (HisPur, Thermo Fisher Scientific) and washed with additional lysis buffer prior to elution with lysis buffer containing 250 mM imidazole. Eluted proteins were then subjected to proteolysis with either 3C protease or SENP1 SUMO protease during overnight 4°C dialysis back to lysis buffer. The cleaved proteins were passed back over cobalt affinity resin, concentrated using 10,000 MWCO centrifugal filters (Amicon, EMD Millipore), and passed over a Superdex 75 pg 16/600 size exclusion column (GE Healthcare) equilibrated in 25 mM Tris, 150 mM NaCl, 5 mM DTT, pH 8.0. Purified protein

was visualized by SDS–PAGE, and appropriate fractions were pooled, concentrated, quantified by absorbance (280 nm), and flash-frozen for storage at −80°C. In the case of ceg7, the SUMO tag was left in place to stabilize the protein.

### Ub activity-based probe assays

The Ub-PA activity-based probe was prepared using intein chemistry as described previously (Wilkinson *et al*, 2005). Activity-based probe reactions were performed as described (Pruneda & Komander, 2019). Bacterial OTUs were prepared at 5 μM concentration in 25 mM Tris, 150 mM NaCl, 10 mM DTT, pH 7.4 and incubated at room temperature for 15 min. Ub-PA was prepared at 7.5 μM concentration in the same buffer. Reactions were initiated by mixing 5 μl each of DUB and Ub-PA, followed by incubation for 1 h at 37°C before quenching in SDS sample buffer. Products were resolved by SDS–PAGE and visualized by Coomassie staining.

### Fluorescence polarization Ub/Ub-like cleavage assays

Fluorescent Ub- and Ub-like-KG(TAMRA) substrates were prepared as described previously (Geurink *et al*, 2012; Basters *et al*, 2014). Cleavage was monitored by fluorescence polarization as previously described (Pruneda & Komander, 2019). Bacterial OTUs were prepared at twice the desired enzyme concentration in 25 mM Tris, 100 mM NaCl, 5 mM β-mercaptoethanol, 0.1 mg/ml BSA, pH 7.4 (FP buffer) and incubated at room temperature for 15 min. Fluorescent Ub/Ub-like substrates were prepared at 20 nM concentration in FP buffer. 5 μl each of DUB and substrate was mixed in a black, low-volume 384-well plate (Greiner), and fluorescence polarization was monitored at room temperature on a Clariostar plate reader equipped with a 540/590 nm filter set (BMG Labtech). Ub/Ubl substrate alone and KG(TAMRA) peptide alone were included as negative and positive controls, respectively, and used to convert polarization values to percent substrate remaining. An increase in fluorescence polarization resulting from noncovalent interaction can result in an apparent substrate remaining value above 100%, as was observed for BurkOTU. To account for FP changes that arise from Ub/Ub-like noncovalent binding or contaminating OTU-independent activity, data from the inactive Cys-to-Ala mutants were used to correct the FP signals in Figs 2C and EV2A. The averages from three technical replicates of one representative assay are shown. Heatmaps display the corrected percent substrate remaining calculated as the average of the final five measurements.

### Ub chain specificity profiling

K27-linked diUb was prepared chemically (van der Heden van Noort *et al*, 2017), M1-linked diUb was expressed and purified as a gene fusion, and the six other linkages were prepared enzymatically (Michel *et al*, 2018). Ub chain cleavage assays were performed as described (Pruneda & Komander, 2019). Bacterial OTUs were prepared at twice the desired concentration in 25 mM Tris, 150 mM NaCl, 10 mM DTT, pH 7.4 and incubated at room temperature for 15 min. diUb chains were prepared at 10 μM in 25 mM Tris, 150 mM NaCl, pH 7.4. The reaction was initiated by mixing 10 μl each of DUB and diUb, and allowed to proceed at 37°C for the indicated time periods. 5 μl reaction samples were

quenched in SDS sample buffer, resolved by SDS–PAGE, and visualized by Coomassie staining. Pixel intensities for the mono- and diUb bands were quantified using ImageJ (Schneider *et al*, 2012) and used to calculate the percent substrate remaining presented in the heatmap.

### Protein crystallization

wMelOTU (1–215) was prepared at 10 mg/ml and crystallized in sitting drop format with 0.2 M sodium acetate, 32% PEG 4K, 0.1 M Tris pH 8.5 at 18°C. Crystals were cryoprotected in mother liquor containing 30% glycerol prior to vitrification.

wMelOTU-Ub was formed by reacting wMelOTU (40–205) with molar excess Ub-C2Br activity-based probe [prepared according to (Wilkinson *et al*, 2005)] at room temperature for 16 h. The covalent wMelOTU-Ub was purified by size exclusion chromatography using a Superdex 75 pg 16/600 column (GE Healthcare). The wMelOTU-Ub complex was prepared at 10 mg/ml and crystallized in sitting drop format with 20% PEG 6K, 0.1 M citrate pH 4.6 at 18°C. Crystals were cryoprotected in mother liquor containing 30% glycerol prior to vitrification.

The EschOTU-Ub complex was formed by reacting EschOTU (184–362) with molar excess His-3C-tagged Ub-C2Br activity-based probe at room temperature for 16 h. The reacted complex was purified using cobalt affinity resin, eluted with 250 mM imidazole, cleaved with 3C protease, and subjected to final purification by size exclusion chromatography using a Superdex 75 pg 16/600 column (GE Healthcare). The EschOTU-Ub complex was prepared at 12 mg/ml and crystallized in sitting drop format with 0.8 M sodium formate, 10% PEG 8K, 10% PEG 1K, 0.1 M sodium acetate pH 4.5 at 18°C. Crystals were cryoprotected in mother liquor containing 25% glycerol prior to vitrification.

### Data collection, structure determination, and refinement

Diffraction data were collected at Diamond Light Source (DLS). Images were integrated using XDS (Kabsch, 2010) or DIALS (Winter *et al*, 2018) software and scaled using Aimless (Evans & Murshudov, 2013). The wMelOTU structure was determined by molecular replacement with Phaser (McCoy *et al*, 2007) using a minimal OTU domain from *Saccharomyces cerevisiae* OTU1 [PDB 3C0R (Messick *et al*, 2008)]. The wMelOTU-Ub structure was determined molecular replacement with Phaser (McCoy *et al*, 2007) using the apo wMelOTU and Ub structures [PDB 1UBQ (Vijay-Kumar *et al*, 1987)] as models. The EschOTU-Ub structure was determined by molecular replacement with Phaser (McCoy *et al*, 2007) using a sieved OTU domain structure generated by MUSTANG-MR with an OTU multiple sequence alignment and set of corresponding structures (Konagurthu *et al*, 2010), in addition to Ub [PDB 1UBQ (Vijay-Kumar *et al*, 1987)]. All structures underwent iterative rounds of manual building in Coot (Emsley *et al*, 2010) and refinement in Phenix (Adams *et al*, 2010). Structure figures were prepared using PyMOL (Schrödinger).

### Comparative OTU structural analysis

The wMelOTU-Ub and EschOTU-Ub crystal structures were compared to published structures from all major OTU subfamilies, including human Otubain, OTUD, OTULIN, and A20 subfamilies as

well as the viral vOTU, PLP2, and PRO subfamilies. A focus was placed on Ub-bound structures that reveal the structural requirements of the S1 binding site. Human Otubain subfamily structures included OTUB1 [PDB 4DDG (Juang *et al*, 2012)] and OTUB2 [PDB 4FJV (Altun *et al*, 2015)]. The human OTUD subfamily included OTUD1 [PDB 4BOP (Mevissen *et al*, 2013)], OTUD2 [PDB 4BOZ (Mevissen *et al*, 2013)], OTUD3 [PDB 4BOU (Mevissen *et al*, 2013)], and OTUD5 [PDB 3TMP (Huang *et al*, 2012)]. Human OTULIN subfamily structures included OTULIN [PDB 3ZNZ (Keusekotten *et al*, 2013)]. Human A20 subfamily structures included A20 [PDB 5LRX (Mevissen *et al*, 2016)], Cezanne [PDB 5LRW (Mevissen *et al*, 2016)], and TRABID [PDB 3ZRH (Licchesi *et al*, 2011)]. The viral vOTU subfamily included CCHFV vOTU [PDB 3PHW (Akutsu *et al*, 2011)], Qalyub virus vOTU [PDB 6DX1 (Dzimianski *et al*, 2019)], Dera Ghazi Khan virus vOTU [PDB 6DX2 (Dzimianski *et al*, 2019)], Taggert virus vOTU [PDB 6DX3 (Dzimianski *et al*, 2019)], and Farallon virus vOTU [PDB 6DX5 (Dzimianski *et al*, 2019)]. The PLP2 and PRO viral subfamily structures included EAV PLP2 [PDB 4IUM (van Kasteren *et al*, 2013)] and TYMV PRO [PDB 4A5U (Lombardi *et al*, 2013)]. Structures were aligned based on their core OTU fold (central β-sheet and two supporting α-helices) and visualized using PyMOL (Schrödinger). During the preparation of this manuscript, a crystal structure of *Legionella* ceg23 was published and we have categorized its predicted S1 site based upon our established framework [PDB 6KS5 (Ma *et al*, 2020)].

## Data availability

Coordinates and structure factors for the wMelOTU, wMelOTU-Ub, and EschOTU-Ub structures have been deposited in the Protein Data Bank (http://www.rcsb.org/) under accession numbers 6W9O, 6W9R, and 6W9S, respectively.

**Expanded View** for this article is available online.

## Acknowledgements
We would like to thank the beamline staff at DLS I04, I04-1, and I24 for their assistance. Access to DLS was supported, in part, by the EU FP7 infrastructure grant BIOSTRUCT-X (contract no. 283570). This work was supported by a VICI grant from the Netherlands Organization for Scientific Research N.W.O. (724.013.002, HO), the Medical Research Council (U105192732, DK), the European Research Council (309756 and 724804, DK), the Lister Institute for Preventative Medicine (DK), an EMBO Long-Term Fellowship (JNP), Oregon Health & Science University (JNP), and The Collins Medical Trust in Portland, OR (JNP). TGF was supported by 5T32GM071338-14, Program in Molecular and Cellular Biosciences.

## Author contributions
DK and KH conceived the project. JNP and DK designed all experiments. KH performed the initial bioinformatic prediction which was manually inspected by JNP and DK. JNP, AFS, TGF, JVN, DJS, and LNM performed biochemical experiments. JVN, PPG, HO, and CGR contributed key reagents. JNP and AFS determined the crystal structures. JNP and DK analyzed the data and wrote the manuscript with input from all of the authors.

## Conflict of interest
The authors declare that they have no conflict of interest.

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
