## [Review Process File · The EMBO Journal]

Identification and characterization of diverse OTU deubiquitinases in bacteria

Alexander F. Schubert, Justine V. Nguyen, Tyler G. Franklin, Paul P. Geurink, Cameron G. Roberts, Daniel J. Sanderson, Lauren N. Miller, Huib Ovaa, Kay Hofmann, Jonathan N. Pruneda and David Komander

Review timeline:

Submission date:	28 th March 2020
Editorial Decision:	27 th April 2020
Revision received:	19 th May 2020
Accepted:	29 th May 2020

Editor: Hartmut Vodermaier

Transaction Report:

1st Editorial Decision

27th April 2020

Thank you for submitting your manuscript on bacterial OTU DUBs for our consideration. Three expert referees have now provided their comments on it, copied below for your information. Given their overall supportive assessment, we would be interested in pursuing the study further for The EMBO Journal, pending adequate revision of a number of specific, mostly presentational, points raised by the reviewers. Any experimental data you may have to address the two main concerns of referee 1 would clearly increase the impact of the work, but at least for the first point regarding possible effector roles of the bacterial OTU DUBs, bioinformatic analyses e.g. of translocation signals would also be helpful. It will furthermore be important to cite and discuss recently published work on *Legionella* Ceg23 (JBC Feb 2020).

REFeree REPORTS

Referee #1:

Schubert, A. et al. predicted a number of OTU deubiquitinases from several classes of pathogenic bacteria through a bioinformatics method. They further confirmed the activities of these deubiquitinases biochemically and determined the substrate specificities for ubiquitin and polyubiquitin. Moreover, they determined the ubiquitin-bound structures of two deubiquitinases they identified and revealed novel modes of substrate recognition by bacterial OTU deubiquitinases. The authors made cross-kingdom analyses of the OTU fold and revealed three variable regions in the S1 substrate recognition site, which set up a framework for OTU deubiquitinase analysis.

This is an interesting study of bacterial deubiquitinases. In bacterial-host interactions, the ubiquitin

pathway is frequently targeted by bacterial effector proteins. Identification of effector deubiquitinases is important for uncovering bacterial pathogenesis. Currently, there is no good strategy to identify and character effector deubiquitinases from numerous bacterial pathogens. This study provides an excellent case for studying bacterial effector deubiquitinases and also provides new insights for the OTU family of deubiquitinases. This paper will be great interesting for the investigators in the field of bacterial-host interactions. I think this paper is suitable for publication in EMBO J.

I have two concerns as following.

1, Except the *Legionella* deubiquitinases Ceg7 and Ceg23, there is no evidence that suggests the other deubiquitinases the authors identified are effector proteins. Bacterial pathogens sometimes encode a pseudo-gene or don't secret the encoded protein. Though it is most likely that these deubiquitinases are indeed effector proteins, the author should make some discussions in the paper. The type III effectors have some conserved signal amino acid in the N-terminus, while the type IV effector usually contains a signal peptide in the C-terminus.

2, The authors examined the substrate specificities of the predicted deubiquitinase via free diUb. But it is unknown whether free polyubiquitin or monoubiquitylated protein as the real substrate of these deubiquitinases. Many intracellular bacterial pathogens encode deubiquitinases to remove ubiquitin moieties from ubiquitylated proteins on their vacuoles in the host cells. Can the authors carry out some assays to discriminate free polyubiquitin or ubiquitylate proteins as the real substrates? Or the authors make some discussion in the paper.

Minor issues:

1, whether Fig 1G is consistent with Fig EV2A?

Referee #2:

In this manuscript, Schubert et al have presented a study aimed at identification and characterization of prokaryotic OTU deubiquitinases (DUBs) in several intracellular bacteria that likely use these as effectors or virulent factors in their interaction with the host. Manipulation of the ubiquitin landscape of the eukaryotic host is emerging as a powerful theme in host-pathogen interaction, first appreciated nearly three decades ago with the discovery of viral factors, such as the E6 protein of the human papilloma virus. Since then a number of bacterial enzymes that mimic the function of eukaryotic E3 ligases, modify ubiquitin and, more recently, completely new ubiquitinating enzymes unrelated to the E1-E2-E3 ubiquitination system of eukaryotes have emerged as key players in host-pathogen interaction. DUBs have also surfaced as important effectors and virulent factors in a wide variety of intracellular bacteria. In a paper published in 2016, Pruneda et al (the same group) had described their elegant work on a diverse group of CE-clan DUBs in bacterial effectors. The current work is built around a similar theme, a broadly similar approach but aimed at a search for the OTU subfamily of DUBs in bacteria. The search has led to demonstration of biochemical DUB activity in a novel group of OTU enzymes and some basic structural principles underlying ubiquitin recognition by these bacterial enzymes.

Prior to this report, the viral OTU DUBs were fairly well known, so there were hints that there would be bacterial OTU counterparts. Indeed, an effector from *Legionella pneumophila* called LotA was the first OTU DUB identified in bacteria (published in 2018). The authors here have asked if, like the CE-clan enzymes, OTU DUBs are present more widely in other intracellular bacteria. In answering this question, the authors have arrived at an interesting discovery of a diverse group of bacterial OTU DUBs in a number of pathogenic species. These DUBs are perhaps a subset of a larger group of enzymes that remain to be identified, and with that the possibility of learning new aspects of host-pathogen interaction.

I think this current paper will be considered as a foundational one on the topic of prokaryotic OTU DUBs and will pave the way for new discoveries in microbiology. The overall presentation in the paper is of high quality, the structural data and other biochemical data have a strong scientific footing, with some insightful analysis. It is therefore my feeling that this paper will be appropriate for the broader readership of the EMBO J.

I would like to the authors to consider the following points which I believe will improve the presentation, the clarity at certain places and the overall impact of the paper.

1. I think the bioinformatics approach leading to the identification of key sequence and structural features of the putative OTU DUBs is interesting. This is not some run-of-the-mill bioinformatics that anyone could have practiced. Considering that, I feel the description given on this topic is somewhat superficial. It could be presented in a flowchart or a scheme showing the distinct steps taken and how that led to the select group of enzymes (including the cut-off used in the selection criteria). For example, how did the results look after the secondary structure analysis using PHYRE2? What is the list of candidates that went into the PHYRE2 search? It would be important to describe this in some details so others can follow a similar scheme in trying to identify novel DUBs.

2. I was surprised to see no mention of LotA appearing in their search. Afterall, it is an established bacterial OTU, that too with two OTU domains.

3. It will help to describe the basis of arriving at the minimal OTU catalytic domain. Some of the constructs have just around 165 residues (such as wMelOTU (40-205)) whereas the BurkOTU construct is much larger (1-506).

Minor Issues:

4. The crystal Structure of Ceg23 has already been published recently. It would be worthwhile including this in the analysis presented in Figure 6. I think it would be among the expanded helical type in Fig 6B.

5. The higher polarization in the case of BurkOTU is intriguing. It has been suggested that it may be indicative of tight binding of the substrate. Does that mean tight binding in an unproductive orientation? Is it possible that the peptide has been cleaved from ubiquitin and the fragment carrying the TAMRA group is associated with the protein? (Running a MALDI ms of the reaction mixture will make this clearer).

6. Please include a discussion on the diverse range of catalytic activity observed with the bacterial OTUs, especially with di-ubiquitin substrates. For example, 100 nM of ceg7 can efficiently cleave diubiquitin substrates within 10 minutes, whereas RickOTU shows similar level of activity only at 10 micromolar enzyme with extended reaction times (Fig EV2 B).

7. In figure 2B, 200% percent substrate remaining does not make much physical sense. Please consider separating that part from the rest as in Figure 1G.

8. RickOTU does not show Ub-PA modification, yet shows some discernible activity with the Ub-TAMRA substrate (Figure 1) and with diubiquitin substrates. Please explain.

Referee #3:

Whereas many OTU-family deubiquitinating enzymes (DUBs) in eukaryotes and viruses have been discovered and characterized, prior to this work by Schubert et al., only a couple of bacterial OTU DUBs were known. Using a bioinformatic screen of bacterial sequences, Schubert et al., have revealed several previously unknown bacterial OTU DUBs. These enzymes are likely to serve roles in bacterial host-pathogen interactions, and the results of biochemical and structural characterization of the new DUBs provides new insight into the possible evolution of this important class of proteases. Importantly, the authors conclude from their analyses of the OTU DUB sequences and 3-dimensional structures of the enzymes' catalytic sites that an 'OTU consensus sequence' cannot necessarily be relied upon for recognition of additional OTU DUBs from other organisms.

Overall, I am impressed with the broad scope of this study and its high technical quality; I have only a few suggestions for minor revisions (see below). Beyond those suggestions, my only other comment is that I'm not convinced of the utility of the authors' variable region (VR) framework

developed to describe the OTU DUBs' S1 ubiquitin binding site (see Fig. 6 and Results beginning at line 333).

1. For many of the results, DUB activities are being compared without specifying their concentrations (e.g., Fig. 2C,F). Especially because different concentrations were used for different enzymes in some experiments (e.g., Fig. 1G), in all cases the concentrations need to be made clear. Along these same lines, in Fig. 2F it would be helpful to indicate whether or not the 8 different enzymes can be compared with each other based on the heat-map color intensities.
2. In Fig. 1, the heat map presented in panel H appears to contradict the results in panel G. Whereas in G the relative wild-type DUB activities are wMelOTU > ceg7 > EschOTU, the heat map indicates EschOTU > wMelOTU > ceg7.
3. Perhaps I missed it - what bacterial sequence database(s) were used to search for the bacterial OTU DUBs? It would be helpful to know where they were NOT found.
4. In Fig. 3D, the models before and after 90{degree sign} rotation are sized differently.

We would like to thank all of the referees and the editorial staff for their continued dedication to peer-reviewed science during these uncertain times. The favorable comments and thoughtful suggestions we received from the referees were very much appreciated, and we have addressed them point-by-point below.

Referee #1:

Schubert, A. et al. predicted a number of OTU deubiquitinases from several classes of pathogenic bacteria through a bioinformatics method. They further confirmed the activities of these deubiquitinases biochemically and determined the substrate specificities for ubiquitin and polyubiquitin. Moreover, they determined the ubiquitin-bound structures of two deubiquitinases they identified and revealed novel modes of substrate recognition by bacterial OTU deubiquitinases. The authors made cross-kingdom analyses of the OTU fold and revealed three variable regions in the S1 substrate recognition site, which set up a framework for OTU deubiquitinase analysis.

This is an interesting study of bacterial deubiquitinases. In bacterial-host interactions, the ubiquitin pathway is frequently targeted by bacterial effector proteins. Identification of effector deubiquitinases is important for uncovering bacterial pathogenesis. Currently, there is no good strategy to identify and character effector deubiquitinases from numerous bacterial pathogens. This study provides an excellent case for studying bacterial effector deubiquitinases and also provides new insights for the OTU family of deubiquitinases. This paper will be great interesting for the investigators in the field of bacterial-host interactions. I think this paper is suitable for publication in EMBO J.

We are happy to hear that the referee appreciated our study and its future impact on the field of bacterial-host interactions.

I have two concerns as following.

1, Except the Legionella deubiquitinases Ceg7 and Ceg23, there is no evidence that suggests the other deubiquitinases the authors identified are effector proteins. Bacterial pathogens sometimes encode a pseudo-gene or don't secret the encoded protein. Though it is most likely that these deubiquitinases are indeed effector proteins, the author should make some discussions in the paper. The type III effectors have some conserved signal amino acid in the N-terminus, while the type IV effector usually contains a signal peptide in the C-terminus.

As the referee indicated, our original prediction that OTU-containing proteins were secreted effectors was based on the premise that their role as deubiquitinases would only be relevant in the context of a host cell. The referee makes a valid point though, and following their suggestion we have performed additional analyses to determine if the OTU-containing proteins we identified are also predicted to have Type III or Type IV secretion signals. We found that all of the bacterial proteins in question were predicted to contain secretion signals, and this analysis is now presented in Extended View Figure 1B and discussed in the Results section.

2, The authors examined the substrate specificities of the predicted deubiquitinase via free diUb. But it is unknown whether free polyubiquitin or monoubiquitylated protein as the real substrate of these deubiquitinases. Many intracellular bacterial pathogens encode deubiquitinases to remove ubiquitin moieties from ubiquitylated proteins on their vacuoles in the host cells. Can the authors carry out some assays to discriminate free polyubiquitin or ubiquitylate proteins as the real substrates? Or the authors make some discussion in the paper.

In the cases where the targets of bacterial DUBs have been identified both *in vitro* and in the context of infection, they appear to match quite well. For example, the *Legionella* effector DUB RavD was shown to specifically cleave Met1-linked Ub chains *in vitro*, and it also removes this signal from the *Legionella*-containing vacuole during infection (Wat et al. 2019 Nat Microbiol). As is the case with some human DUBs, however, it is also possible that bacterial DUBs have evolved to be highly substrate specific. In that event it is likely that no proxy substrate for *in vitro* reactions would be a suitable mimic. We have added additional discussion on these possibilities to the Discussion section so that the limitations of our substrate specificity analyses are clear.

Minor issues:

1, whether Fig 1G is consistent with Fig EV2A?

Figure 1G reports DUB activities of the tested bacterial OTUs at various enzyme concentrations, while Figure EV2A reports the activity of EschOTU at 50 nM (half that used in Figure 1G) against Ub- and Ub-like substrates. The only other difference is that the data in Figure EV2A have been corrected by subtracting data collected with the catalytically inactive CA mutant.

Referee #2:

In this manuscript, Schubert et al have presented a study aimed at identification and characterization of prokaryotic OTU deubiquitinases (DUBs) in several intracellular bacteria that likely use these as effectors or virulent factors in their interaction with the host. Manipulation of the ubiquitin landscape of the eukaryotic host is emerging as a powerful theme in host-pathogen interaction, first appreciated nearly three decades ago with the discovery of viral factors, such as the E6 protein of the human papilloma virus. Since then a number of bacterial enzymes that mimic the function of eukaryotic E3 ligases, modify ubiquitin and, more recently, completely new ubiquitinating enzymes unrelated to the E1-E2-E3 ubiquitination system of eukaryotes have emerged as key players in host-pathogen interaction. DUBs have also surfaced as important effectors and virulent factors in a wide variety of intracellular bacteria. In a paper published in 2016, Pruneda et al (the same group) had described their elegant work on a diverse group of CE-clan DUBs in bacterial effectors. The current work is built around a similar theme, a broadly similar approach but aimed at a search for the OTU subfamily of DUBs in bacteria. The search has led to demonstration of biochemical DUB activity in a novel group of OTU enzymes and some basic structural principles underlying ubiquitin recognition by these bacterial enzymes.

Prior to this report, the viral OTU DUBs were fairly well known, so there were hints that there would be bacterial OTU counterparts. Indeed, an effector from *Legionella pneumophila* called LotA was the first OTU DUB identified in bacteria (published in 2018). The authors here have asked if, like the CE-clan enzymes, OTU DUBs are present more widely in other intracellular bacteria. In answering this question, the authors have arrived at an interesting discovery of a diverse group of bacterial OTU DUBs in a number of pathogenic species. These DUBs are perhaps a subset of a larger group of enzymes that remain to be identified, and with that the possibility of learning new aspects of host-pathogen interaction.

I think this current paper will be considered as a foundational one on the topic of prokaryotic OTU DUBs and will pave the way for new discoveries in microbiology. The overall presentation in the paper is of high quality, the structural data and other biochemical data have a strong scientific footing, with some insightful analysis. It is therefore my feeling that this paper will be appropriate for the broader readership of the EMBO J.

We would like to thank the reviewer for their kind words and for their appreciation of how this work will contribute to the fields of prokaryotic DUBs and microbiology as a whole.

I would like to the authors to consider the following points which I believe will improve the presentation, the clarity at certain places and the overall impact of the paper.

1. I think the bioinformatics approach leading to the identification of key sequence and structural features of the putative OTU DUBs is interesting. This is not some run-of-the-mill bioinformatics that anyone could have practiced. Considering that, I feel the description given on this topic is somewhat superficial. It could be presented in a flowchart or a scheme showing the distinct steps taken and how that led to the select group of enzymes (including the cut-off used in the selection criteria). For example, how did the results look after the secondary structure analysis using PHYRE2? What is the list of candidates that went into the PHYRE2 search? It would be important to describe this in some details so others can follow a similar scheme in trying to identify novel DUBs.

To provide additional detail, we have added a workflow describing our approach to the prediction and curation of bacterial OTUs to Fig EV1A. The workflow presents our iterative refinement of multiple sequence alignments and profile HMMs. Since we first began our study, this method has become a more commonplace approach, and similar methods are integrated into online tools such as PSI-BLAST or JackHMMER. Our approach to manual curation is also described in the workflow, and representative output for one of the predicted OTUs is shown. The new workflow is now referred to in the Results section as we present our predicted bacterial OTUs.

2. I was surprised to see no mention of LotA appearing in their search. After all, it is an established bacterial OTU, that too with two OTU domains.

We agree, but it is unclear to us why LotA was not detected alongside our other predictions. In the Results section, we now discuss that LotA was not detected, despite apparently having two OTU domains.

3. It will help to describe the basis of arriving at the minimal OTU catalytic domain. Some of the constructs have just around 165 residues (such as wMelOTU (40-205)) whereas the BurkOTU construct is much larger (1-506).

To address this, we have clarified our approach to construct design by providing additional detail in the Methods section. The design was focused on obtaining minimal OTU domains based on active site and secondary structure prediction. In the case of BurkOTU, including an additional structured region N-terminal to the OTU domain was necessary to obtain soluble protein expression.

Minor Issues:

4. The crystal Structure of Ceg23 has already been published recently. It would be worthwhile including this in the analysis presented in Figure 6. I think it would be among the expanded helical type in Fig 6B.

Thank you for your suggestion. We have now included a reference to the recent crystal structure of ceg23 as validation of our predicted active site topology described in Figure 5. We have also analyzed the predicted S1 site of ceg23 using our established framework and

categorized its variable regions in Figure 6.

5. The higher polarization in the case of BurkOTU is intriguing. It has been suggested that it may be indicative of tight binding of the substrate. Does that mean tight binding in an unproductive orientation? Is it possible that the peptide has been cleaved from ubiquitin and the fragment carrying the TAMRA group is associated with the protein? (Running a MALDI ms of the reaction mixture will make this clearer).

This is an interesting phenomenon and we thank the reviewer for their suggestion. We believe that the unproductive binding observed in the fluorescence polarization assay could indicate either the presence of a high-affinity binding site outside of the S1 site (perhaps an S2 site or additional Ub binding site as observed in other DUB families (e.g. Berk et al. Nat. Commun. 2020)), or that binding does occur in the S1 site but the BurkOTU catalytic triad must be oriented by some other means. Both hypotheses are consistent with the observation that BurkOTU can cleave diUb chains, i.e. binding might be driven by a high affinity S1' site or binding in the S1' site could serve to activate the catalytic triad (similar to the human DUB OTULIN). We have added additional text in the Results section to explain these possibilities.

The alternative explanation suggested by the reviewer was a good one, but for several reasons we believe that this is an unlikely possibility. First, we observe the same increase in fluorescence polarization with the catalytically inactive CA mutant of BurkOTU (Fig. 2B), indicating that cleavage of the Ub-modified fluorescent peptide is not required for its noncovalent association. Second, in work that we have not included, we can observe the same signs of a strong noncovalent interaction with a different fluorescent Ub construct that instead carries a noncleavable C-terminal FIAsh tag.

6. Please include a discussion on the diverse range of catalytic activity observed with the bacterial OTUs, especially with di-ubiquitin substrates. For example, 100 nM of ceg7 can efficiently cleave diubiquitin substrates within 10 minutes, whereas RickOTU shows similar level of activity only at 10 micromolar enzyme with extended reaction times (Fig EV2 B).

As the reviewer points out we do see a wide range of activities across the bacterial OTUs. This trend is similar to that observed among the human OTUs (see Mevissen et al. 2013 Cell) and could reflect the many alterations of the S1 binding site. We now highlight this point in the Discussion section.

7. In figure 2B, 200% percent substrate remaining does not make much physical sense. Please consider separating that part from the rest as in Figure 1G.

We agree that 200% substrate does not make physical sense, but we felt this was a reasonable compromise to ensure that all of the data are presented in the same format. To account for this, we have done our best to explain how noncovalent interaction can produce an apparent substrate remaining value above 100% in the Results, Figure Legends, and Methods sections. As the reviewer has suggested, we have also broken the y-axis in Fig 2B so that the presentation is more consistent with Fig 1G.

8. RickOTU does not show Ub-PA modification, yet shows some discernible activity with the Ub-TAMRA substrate (Figure 1) and with diubiquitin substrates. Please explain.

This was a peculiar observation but not entirely surprising for us. Anecdotally, we have found that some DUBs prefer different types of electrophilic warhead at the Ub C-terminus, and while

we find that the Ub-PA can react with most examples it is not universally perfect. We routinely test several measures of DUB activity to account for examples like RickOTU. For example, in our early characterization of OTULIN (Keusekotten et al. Cell 2013), we noted no reactivity with conventional Ub activity-based probes but strong hydrolysis of diUb chains.

Referee #3:

Whereas many OTU-family deubiquitinating enzymes (DUBs) in eukaryotes and viruses have been discovered and characterized, prior to this work by Schubert et al., only a couple of bacterial OTU DUBs were known. Using a bioinformatic screen of bacterial sequences, Schubert et al., have revealed several previously unknown bacterial OTU DUBs. These enzymes are likely to serve roles in bacterial host-pathogen interactions, and the results of biochemical and structural characterization of the new DUBs provides new insight into the possible evolution of this important class of proteases. Importantly, the authors conclude from their analyses of the OTU DUB sequences and 3-dimensional structures of the enzymes' catalytic sites that an 'OTU consensus sequence' cannot necessarily be relied upon for recognition of additional OTU DUBs from other organisms.

Overall, I am impressed with the broad scope of this study and its high technical quality; I have only a few suggestions for minor revisions (see below). Beyond those suggestions, my only other comment is that I'm not convinced of the utility of the authors' variable region (VR) framework developed to describe the OTU DUBs' S1 ubiquitin binding site (see Fig. 6 and Results beginning at line 333).

We thank the reviewer for their appreciation of the scope and quality of our work. Regarding the utility of the analysis we have performed on the S1 site, perhaps the reviewer would be interested to see that a recent crystal structure of the *Legionella* ceg23 DUB has been determined (Ma et al. JBC 2020) and in our revised manuscript we have categorized its predicted S1 site into our existing framework.

1. For many of the results, DUB activities are being compared without specifying their concentrations (e.g., Fig. 2C,F). Especially because different concentrations were used for different enzymes in some experiments (e.g., Fig. 1G), in all cases the concentrations need to be made clear. Along these same lines, in Fig. 2F it would be helpful to indicate whether or not the 8 different enzymes can be compared with each other based on the heat-map color intensities.

As the reviewer indicated, the enzyme concentrations of each bOTU were adjusted to suit the amount of activity we observed, and thus most of our comparisons are made within a bOTU (e.g. Ub/Ub-like specificity or Ub chain specificity). To make this clear, we have followed the reviewer's recommendation and included the enzyme concentration in each figure.

2. In Fig. 1, the heat map presented in panel H appears to contradict the results in panel G. Whereas in G the relative wild-type DUB activities are wMelOTU > ceg7 > EschOTU, the heat map indicates EschOTU > wMelOTU > ceg7.

We initially calculated the values reported in the heap map of Fig. 1H by comparing the initial and final time points of each assay. Performing the comparison in this manner corrected for the increase in fluorescence polarization we observed with BurkOTU. Unfortunately, some of the wild-type enzymes (namely ceg7) had already cleaved an appreciable amount of substrate by the first time point, and this led to the slight differences noted between Fig. 1G and H. In the

revised Fig. 1H, we now compare the final time points to initial measurements made in the equivalent assay performed with the catalytically inactive enzyme. This way, the increase observed with BurkOTU can still be accounted for and we get a more accurate representation of the substrate remaining after the reaction.

3. Perhaps I missed it - what bacterial sequence database(s) were used to search for the bacterial OTU DUBs? It would be helpful to know where they were NOT found.

We performed our search in the UniProt database. This detail is included in the Methods section and now made clearer in the new Fig. EV1A that presents a workflow of our bioinformatic prediction process.

4. In Fig. 3D, the models before and after 90{degree sign} rotation are sized differently.

We have double-checked Fig. 3D and believe that the apparent size difference is an optical illusion resulting from an elongated structure being viewed end-on (left image) and along its longer axis (right image).

Thank you for submitting your revised manuscript for our consideration. We have now assessed your responses and modifications answering to the referees' comments, and are pleased to inform you that we have now accepted the study for publication in The EMBO Journal.

Corresponding Author Name: David Komander

Journal Submitted to: The EMBO Journal

Manuscript Number: EMBOJ-2020-105127